# Learning enhances behaviorally relevant representations in apical dendrites

Sam E Benezra[1,2], Kripa B Patel[2,3], Citlali Perez Campos[2,3], Elizabeth MC Hillman[1,2,3], Randy M Bruno[1,2,4,5]*

[1]Department of Neuroscience, Columbia University, New York, United States; [2]Kavli Institute for Brain Science, Columbia University, New York, United States; [3]Departments of Biomedical Engineering and Radiology, Columbia University, New York, United States; [4]Zuckerman Mind Brain Behavior Institute, Columbia University, New York, United States; [5]Department of Physiology, Anatomy & Genetics, University of Oxford, Oxford, United Kingdom

## eLife Assessment

This **important** study uses calcium imaging to show an increase in the selectivity of the sensory-evoked response in the apical dendritic tuft of layer 5 barrel cortex neurons as mice learn a whisker-dependent discrimination task. The evidence supporting the conclusions is **compelling**, and this work will be of great interest to neuroscientists working on reward-based learning and sensory processing.

*For correspondence:
randy.bruno@dpag.ox.ac.uk

## Abstract

Learning alters cortical representations and improves perception. Apical tuft dendrites in cortical layer 1, which are unique in their connectivity and biophysical properties, may be a key site of learning-induced plasticity. We used both two-photon and SCAPE microscopy to longitudinally track tuft-wide calcium spikes in apical dendrites of layer 5 pyramidal neurons in barrel cortex as mice learned a tactile behavior. Mice were trained to discriminate two orthogonal directions of whisker stimulation. Reinforcement learning, but not repeated stimulus exposure, enhanced tuft selectivity for both directions equally, even though only one was associated with reward. Selective tufts emerged from initially unresponsive or low-selectivity populations. Animal movement and choice did not account for changes in stimulus selectivity. Enhanced selectivity persisted even after rewards were removed and animals ceased performing the task. We conclude that learning produces long-lasting realignment of apical dendrite tuft responses to behaviorally relevant dimensions of a task.

## Introduction

Learning and memory depend on the ability of biological networks to alter their activity based on past experience. For example, as animals learn the behavioral relevance of stimuli in a sensory discrimination task, neural representations of those stimuli are enhanced (*Poort et al., 2015*; *Liu et al., 2020*; *Henschke et al., 2020*; *Beitel et al., 2003*; *Goltstein et al., 2013*; *David et al., 2012*; *Fritz et al., 2003*), potentially improving the salience of information relayed to downstream areas. Studies in primary somatosensory (S1) (*Banerjee et al., 2020*) and visual cortex (*Liu et al., 2020*) have revealed that top-down signals from distant cortical regions can modify sensory representations during learning, although the cellular and circuit mechanisms underlying this plasticity remain unclear.

Cortical layer 1, comprised mainly of apical tuft dendrites of layer 5 (L5) and layer 2/3 pyramidal neurons, may be a key site driving the enhancement of sensory representations during learning. Apical

tufts are anatomically well positioned for learning, receiving top-down signals from numerous cortical and thalamic areas (*Zhang and Bruno, 2019*; *Rubio-Garrido et al., 2009*; *Cauller et al., 1998*). While L5 distal tufts are electrically remote and far from the soma, they are in close proximity to the highly electrogenic calcium spike initiation zone at the main bifurcation of the apical dendrite, and form a separate biophysical and processing compartment from the proximal dendrites (*Amitai et al., 1993*; *Yuste et al., 1994*; *Schiller et al., 1997*; *Larkum et al., 2009*; *Sandler et al., 2016*). Top-down signals arriving at the tuft can trigger tuft-wide dendritic calcium spikes in L5 neurons (*Manita et al., 2015*), which can modulate synaptic plasticity across the entire dendritic tree (*Roelfsema and Holtmaat, 2018*) and potently drive somatic burst firing (*Larkum et al., 2009*; *Larkum and Zhu, 2002*; *Larkum et al., 2004*; *Schwindt and Crill, 1999*; *Larkum et al., 2001*; *Manita et al., 2017*). Consistent with this observation, L5 apical dendrite activity is highly correlated with somatic activity (*Francioni et al., 2019*; *Beaulieu-Laroche et al., 2019*). Therefore, by strongly influencing somatic activity, L5 apical dendritic calcium spikes can play an important role in modulating cortical output. Several neuromodulators can augment the excitability of the apical tuft and increase the likelihood of eliciting calcium spikes (*Labarrera et al., 2018*; *Brombas et al., 2014*), which could be a substrate for control of plasticity by behavioral state. Consistent with these ideas, we recently demonstrated that during behavioral training with positive reinforcements, apical tufts in sensory cortex acquire associations that extend beyond their normal sensory modality (*Lacefield et al., 2019*). In mouse models of dementia and Alzheimer's disease (*Luebke et al., 2010*; *Tsai et al., 2004*), tuft dendrites exhibit degeneration which may contribute to the cognitive and memory deficits.

L5 pyramidal neurons are the major source of output from cortex, targeting numerous subcortical structures that affect behavior. The activity of apical dendrites is known to correlate with stimulus intensity, and manipulating L5 apical dendrites and their inputs impacts performance of sensory tasks (*Manita et al., 2015*; *Xu et al., 2012*; *Takahashi et al., 2020*; *Takahashi et al., 2016*). Apical dendritic calcium spikes of pyramidal cells could be a crucial cellular mechanism in learning-related plasticity and behavioral modification (*Roelfsema and Holtmaat, 2018*; *Bittner et al., 2017*; *Doron et al., 2020*). However, sensory representations of apical tufts, as well as possible changes across learning, have received little attention.

To address this question, we used two-photon microscopy and a high-speed volumetric imaging technique called Swept Confocally-Aligned Planar Excitation (SCAPE; *Bouchard et al., 2015*; *Hillman et al., 2018*) to longitudinally track the activity of GCaMP6f-expressing L5 apical tufts in barrel cortex during a sensory discrimination task. We found that apical tufts underwent extensive dynamic changes in selectivity for task-relevant stimuli as performance improved, even though only one of the stimuli was unrewarded. These changes in responses persisted even after animals disengaged from the task, demonstrating that learning induced long-lasting changes in tuft sensory representations. Animals that were exposed to the same stimulation protocol without any reinforcement did not develop enhanced representations. Our results show for the first time that reinforcement learning expands apical tuft sensory representations along behaviorally relevant dimensions.

## Results

### Direction discrimination behavior

We devised an awake head-fixed mouse conditioning paradigm that enables controlled investigation of reinforcement effects across learning (*Figure 1a and b*). In addition to discriminating tactile objects, rodents are known to sense wind direction using their whiskers (*Yu et al., 2016a*; *Yu et al., 2016b*) and can be trained to discriminate different directions of whisker deflections (*Nakamura et al., 2009*; *Bernhard et al., 2020*). With this in mind, we directed brief (100 ms) air puffs at the whiskers in either of two directions: rostrocaudal (backward) or ventrodorsal (upward). One of the directions was paired with a water reward delivered 500 ms after the air puff and thus constituted a conditioned stimulus (CS+). No reward was given for the other direction (CS-).

Licking and whisking were monitored throughout the session (*Figure 1c and d*). Stimuli elicited a brief passive whisker deflection followed by active whisking over the subsequent ~1.5 s (analyzed below, Figure 6). Any anticipatory licks prior to reward delivery were counted as a response. Typically, on the first session, mice exhibited few anticipatory licks to either stimulus (*Figure 1c*, top, grey shading). By session 2 or 3, mice had learned an association between whisker deflection and reward,

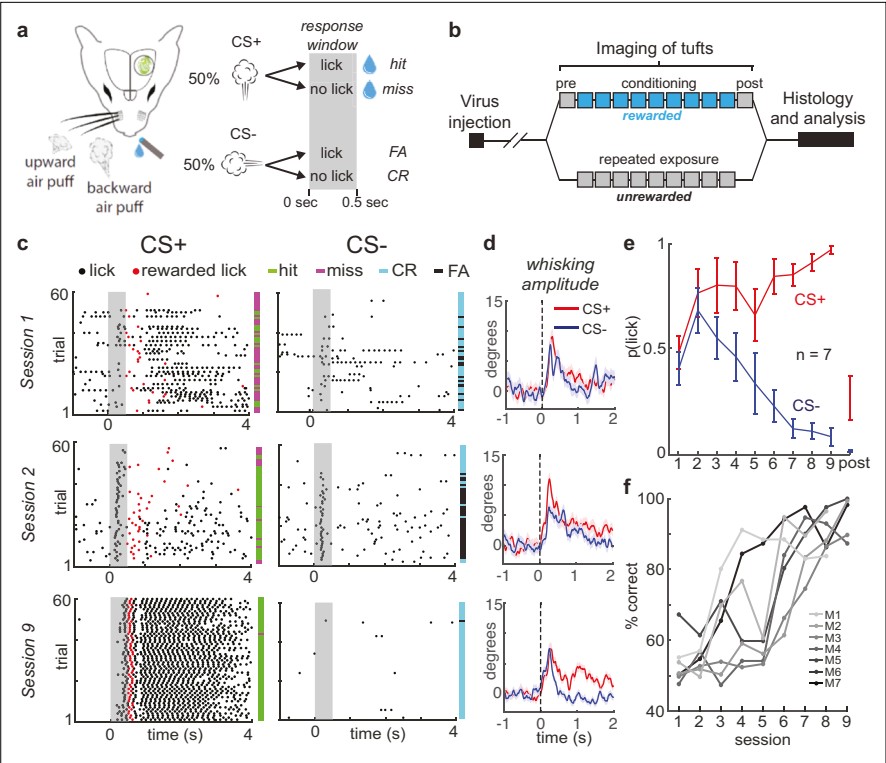

**Figure 1.** Mice rapidly learn to discriminate stimulus direction in head-fixed paradigm. (**a**) A water droplet is paired with air puffs in one direction (CS+) but not the other (CS-). Licking in anticipation of water is assessed in the response window just after CS+ or CS- and prior to water delivery for the CS+ (grey bar). (**b**) Experimental timeline. 2–3 weeks after virus injection, naïve tuft responses to stimuli are recorded (pre). The CS+ is then paired with water for 8–9 days (blue). On the last day, stimuli are presented without reward (post). In a separate group of mice, the same stimuli are presented over 9 days in the absence of reward (unrewarded group). (**c**) Lick rasters for three different sessions in one example mouse. On session 9, the CS+ but not the CS- reliably elicits licks. (**d**) Mean baseline-subtracted whisking amplitude aligned to the CS+ (red) and CS- (navy) across sessions 1, 2, and 9 of an example mouse. (**e**) Learning curve demonstrates rapid learning. Mean probability of at least one lick in the response window across sessions. (**f**) Behavioral performance of each mouse in the rewarded group (M1 – M7).

but could not discriminate the CS+ and CS- (middle). Within a week (by sessions 7–9), every mouse we tested learned to reliably lick to the CS+ while withholding licks to the CS-, performing substantially above chance after a single week of training (*Figure 1c*, bottom; *Figure 1e and f*). Thus, mice rapidly learned to discriminate the direction of whisker stimuli in our behavioral task.

## Overall stimulus-evoked activity is unbiased and stable across conditioning

To investigate the effects of reinforcement learning on apical tuft activity, we imaged apical tufts (433x433 µm field of view) across conditioning days as well as on an unrewarded pre-conditioning day to measure naïve stimulus responses and an unrewarded post-conditioning day to detect any long-lasting changes in responses (*Figure 1b*). Mice remained water-restricted on the post-conditioning day and continued licking for reward toward the beginning of the session (see below). We virally delivered the gene for Cre-dependent GCaMP6f (*Chen et al., 2013*) in the barrel cortex of Rbp4-Cre mice, which labels a heterogeneous population of pyramidal neurons comprising approximately 50% of layer 5 (*Lacefield et al., 2019*; *Kozorovitskiy et al., 2012*; *Glickfeld et al., 2013*). By targeting our injections to layer 5B, we predominantly labeled thick-tufted pyramidal neurons (see Methods). Using intrinsic signal imaging, we mapped the location of the C2, D2, and gamma whisker barrel columns and identified an overlapping region in layer 1 with sufficient GCaMP6f expression (*Figure 2a*). The air puff nozzles were aimed toward the whiskers corresponding to this region. Dendritic activity was

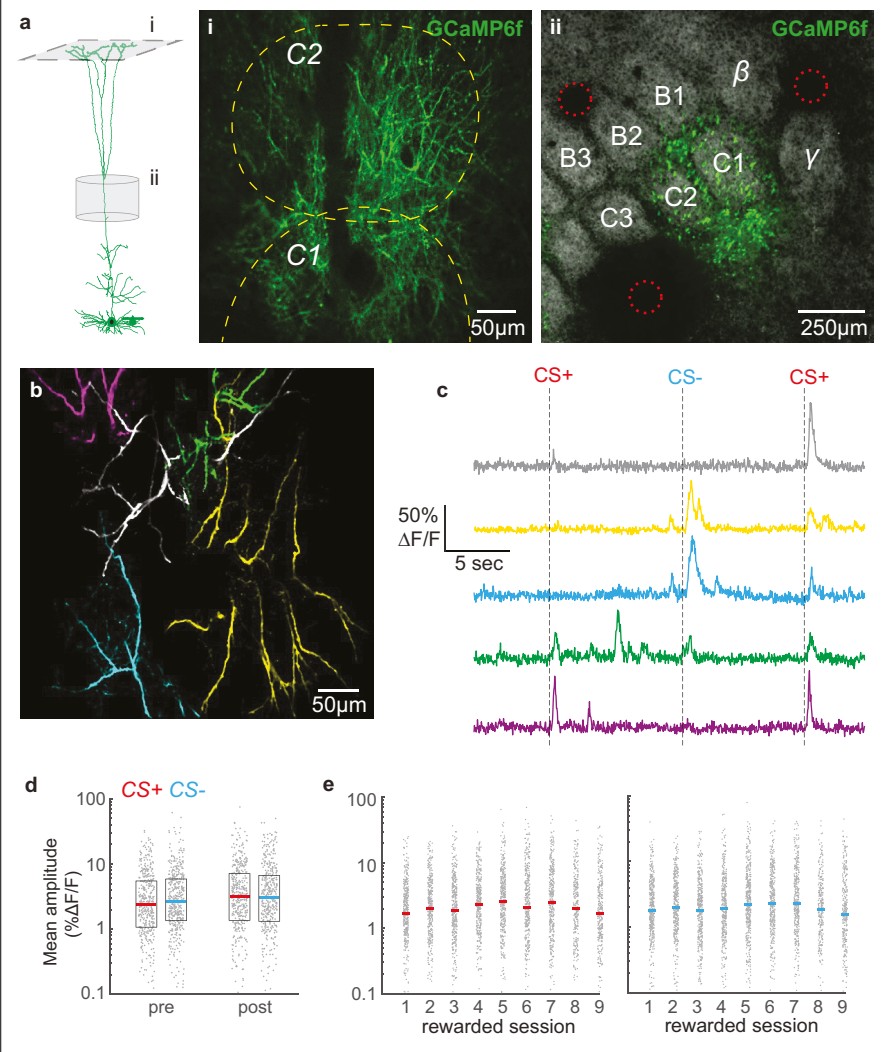

**Figure 2.** Overall tuft response to stimuli is unbiased and relatively stable across conditioning. (**a**) Dendritic activity was recorded in layer 1 (**i**) in the C1/C2 barrel columns (**ii**). (**i**) Two-photon image ~60 μm deep relative to pia. Dashed yellow lines denote C1 and C2 boundaries from intrinsic imaging. Single cell reconstruction in left panel from *Ramirez et al., 2014*. (**ii**) Tangential section through layer 4 showing barrels stained with streptavidin-Alexa 647 and GCaMP6f-expressing apical trunks. Red circles indicate location of 2-photon lesions to mark the imaging region for post-hoc analysis. (**b**) Overlay of five segmented pseudo-colored tufts from imaging field in panel A(i). (**c**) Time courses of calcium responses of example tufts in panel b to three air puffs (dashes). (**d**) Amplitude for CS+ (red) and CS- responses (blue), computed for each segmented tuft in the first 1.5 s post-stimulus (grey points), do not differ within or across sessions. Colored lines indicate median. (**e**) Similar to panel d, but showing data for all conditioning sessions.

The online version of this article includes the following figure supplement(s) for figure 2:

**Figure supplement 1.** CS +trials evoke a second, long-latency peak during early learning, but not late learning.

longitudinally recorded from the same field-of-view (horizontal location and depth) in layer 1 across all sessions (*Video 1*).

To extract calcium signals from individual cells, we segmented tufts using CaImAn, a sparse non-negative matrix factorization method that clusters pixels according to their temporal correlation (*Giovannucci et al., 2019*; see Methods), and analyzed regions of interest exhibiting apical tuft structure (*Figure 2b*; 65±15 tufts per mouse; mean ± SD). Individual segmented tufts were substantial in their spatial extent (>100 μm), reflecting tuft-wide voltage-gated calcium spikes rather than branch-specific N-methyl-D-aspartate (NMDA) receptor-mediated spikes. All calcium analyses hereafter refer to tuft-wide calcium spikes. Average responses to an event include failures. In many tufts, the CS+

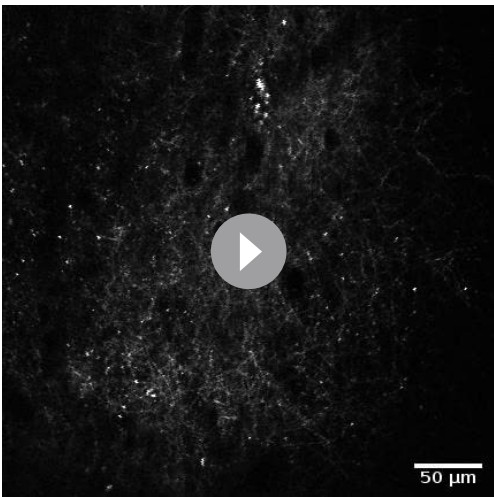

**Video 1.** Example two-photon microscopy movie during behavioral session. Playback speed is in real time. 'CS+' and 'CS-' denote times of stimulus onset. 433 x 433 µm field of view.

https://elifesciences.org/articles/98349/figures#video1

and CS- reliably evoked an influx of calcium that robustly activated the tuft (examples in *Figure 2c*). Successful calcium events across tufts averaged 28% ΔF/F, consistent with previous studies of layer 5 apical dendrites (*Manita et al., 2015*; *Xu et al., 2012*). Interestingly, during intermediate but not early learning, the average population response to the CS+ exhibited a two-peak structure (*Figure 2—figure supplement 1*, session 4) similar to tuft reward-related signals we observed previously in barrel cortex (*Lacefield et al., 2019*). By the last-rewarded and post sessions, the second CS+ peak was no longer visible, which could be an endpoint of mice learning that the conditioned stimulus predicts the upcoming reward.

Reward can alter somatic receptive fields in the auditory, visual, and somatosensory cortex of both rodents and non-human primates such that rewarded stimulus representations become more robust after learning (*Beitel et al., 2003*; *Goltstein et al., 2013*; *Lacefield et al., 2019*; *Pakan et al., 2018*), although cortical sensory responses can remain unchanged during learning (*Wang et al., 2020*). We investigated whether calcium responses to the CS+ increased in the tuft population as animals learned its association with reward (*Figure 2*). Average responses of tufts to the CS+ and CS- were similar during the pre-conditioning session (*Figure 2d*; p=0.20, signed rank test, n=440 pre tufts and 418 post tufts), indicating that there was no inherent bias in the population toward a particular stimulus in naïve animals. Surprisingly, even after learning, responses to the CS+ and CS- were similar on the last- and post-conditioning sessions (p=0.62, 0.64, respectively, signed rank test, *Figure 2d and e*), revealing that no bias develops for the CS+ among dendritic tufts. Only a minority of tufts exhibited statistically significant (see Methods) average responses to air puff stimuli (CS+ responsive: 26 ± 8%; CS- responsive: 25 ± 8%; mean ± SD across all sessions). When we excluded responses that were not statistically significant, we again found no difference between the average response amplitudes to the CS+ and CS- on the pre, last-rewarded, and post sessions (p=0.65, 0.31, and 0.69, respectively, rank sum test; data not shown). Similarly, the probability of transients in response to CS+ versus CS- (see Methods) did not differ during pre-conditioning or post-conditioning sessions (p=0.66 and p=0.44, respectively, data not shown). Therefore, reinforcement learning in our paradigm does not bias tuft representations toward the rewarded stimulus.

While a bias for the CS+ did not develop after learning, we wondered whether overall tuft responses to both conditioned stimuli increased as animals learned the task. Linear regression analysis revealed that conditioning session number was a poor predictor of both CS+ and CS- amplitudes (All tufts $R^2$, CS+: 0.0064, CS-: 0.0035, *Figure 2e*; Significantly responding tufts $R^2$, CS+: 0.014, CS-: 0.014, data not shown). We did find a small but significant decrease in amplitude from pre to last for CS+ (p<0.01) and CS- (p<$10^{-7}$), but this was not permanent: amplitudes did not significantly differ between the pre and post sessions (*Figure 2d*; p=0.53, 0.33, CS+ and CS- respectively, Wilcoxon rank sum test). Taken together, these findings demonstrate that reinforcement learning does not robustly bias the magnitudes of tuft calcium responses to either stimulus at the population level.

## Development of tuft selectivity with task learning

While learning produced no bias in overall tuft activity, learning might enhance selectivity for conditioned stimuli. Barrel cortex neurons are tuned to the angle of whisker deflection (*Bruno et al., 2003*; *Bruno and Sakmann, 2006*; *Ramirez et al., 2014*), indicating that the sets of synaptic connections activated by the CS+ and CS- may be overlapping but should not be identical. Therefore, the possibility exists that responses to the CS+ and CS- can change independently of each other. To examine

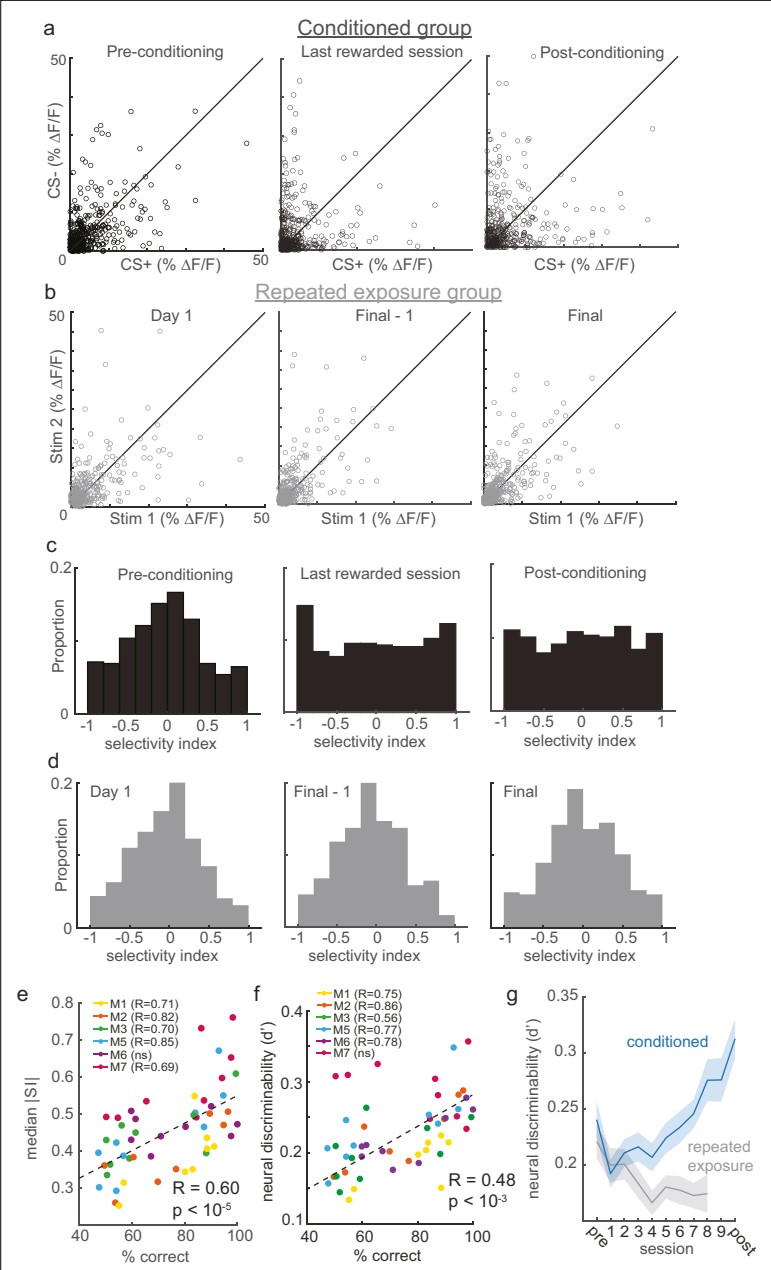

**Figure 3.** Reinforcement learning, but not stimulus exposure, enhances tuft selectivity for CS+ and CS- stimuli. (**a**) Across the indicated sessions, individual tufts (circles) exhibit larger biases to CS+ or CS- (pooled across all conditioned mice). (**b**) Repeated exposure to stimuli does not bias individual tufts to CS+ or CS-. (**c**) Conditioning reshapes distribution of selectivity indices for tufts from Normal on pre-conditioning session to uniform on post-conditioning session. (**d**) Distribution of tuft selectivity indices remains Normally distributed throughout all repeated exposure sessions. (**e**) Selectivity (median SI magnitude of tufts for each session) increases with behavioral performance of six animals. (**f**) Same as panel e, but with neural discriminability plotted on the y axis. (**g**) Neural discriminability (mean ±sem) of tufts, pooled across all animals on each session, increases with conditioning and decreases with repeated exposure.

The online version of this article includes the following figure supplement(s) for figure 3:

**Figure supplement 1.** Selectivity was enhanced in individual animals that received rewards.

**Figure supplement 2.** Decoder performance improves after conditioning, but not repeated exposure.

this, we compared the amplitude of the average response to CS+ and CS- trials for all segmented tufts on the pre, last-rewarded, and post sessions (*Figure 3a*; n=7 mice; 465 pre, 442 last-rewarded, and 430 post tufts). In agreement with our previous analysis, we found no significant bias in response amplitude toward CS+ or CS- during any of the three sessions (*Figure 3a*; Pre: p=0.20; last-rewarded: p=0.43; Post: p=0.64, sign-rank test). Under naive conditions during the pre session, most tufts that responded to air puff stimuli did not strongly prefer the CS+ or CS- (*Figure 3a*, left). Surprisingly, on the last-rewarded session and the unrewarded post-conditioning session, we observed a prominent shift in the response distribution, where many tufts exhibited more selective responses to one stimulus or the other (*Figure 3a*, middle and right).

Plasticity can occur after repeated exposure to stimuli even in the absence of reinforcements (*Yao and Dan, 2001*; *Dragoi et al., 2002*; *Dragoi et al., 2000*; *Zhang et al., 2015*; *Chu et al., 2016*). To test whether enhanced selectivity depended on reinforcement, we imaged a separate group of similarly water-restricted mice that were repeatedly exposed to the same stimuli for the same number of days but without any reward. These mice only received water in their home cage following each imaging session, but never during stimulus presentation. Repeated exposure mice exhibited a stable distribution of response selectivity over time (*Figure 3b*; a separate cohort of 7 mice; 317, 313, and 321 tufts on Day 1, Day 8, and Day 9, respectively). These results suggest that reinforcement learning, and not simply repeated stimulus exposure, drives apical tufts to become more selective for either the CS+ or CS-.

To directly quantify the response selectivity of tufts, we computed a selectivity index (SI; see Methods) ranging from –1 (exclusively CS- responsive) to 1 (exclusively CS+ responsive) for each tuft. Initially in both the conditioned and repeated exposure mice, the SI distribution was centered around zero, indicating that most tufts in naïve animals did not strongly prefer either stimulus (*Figure 3c and d*, left panels). Consistent with our other analyses (*Figure 2d*), the mean SI remained close to zero for each of the three sessions (*Figure 3—figure supplement 1d*; −0.049,−0.001, and 0.003 for pre-conditioning, last rewarded, and post-conditioning days, respectively; one-way ANOVA p=0.37), confirming that learning produced no overall bias toward one particular stimulus among the population. During learning, the SI distribution of conditioned but not repeated exposure mice shifted markedly, whereby a much greater proportion of neurons were highly selective for either the CS+ or CS- (*Figure 3c and d*, middle and right panels, |SI| pre versus last-rewarded: $p<10^{-6}$, |SI| pre versus post: $p<10^{-5}$; Wilcoxon rank sum test). These effects can even be observed within individual mice (*Figure 3—figure supplement 1*). Notably, different tufts within the same animal exhibited opposite changes in selectivity (*Figure 3—figure supplement 1a and b*). Learning significantly increased tuft selectivity in individual conditioned mice, but not repeated exposure mice (*Figure 3—figure supplement 1c*). The degree of enhancement in tuft selectivity was closely correlated with conditioned animals' ability to discriminate stimuli across sessions (*Figure 3e*; Pearson's r=0.60, $p<10^{-5}$).

Whereas selectivity magnitude (|SI|) only considers the amplitude of tuft responses to CS+ and CS-, their discriminability also depends on their variability. For example, a large difference in CS+ and CS- responses would not be discriminable if the variability of those responses were very high; a small difference might be discriminable if the variability were low. We therefore additionally calculated a d-prime metric of neural discriminability that normalizes differences in response magnitudes to each stimulus by their variability (see Methods). Similar to selectivity magnitude, we found that neural discriminability was correlated with behavioral performance (*Figure 3f*). In conditioned animals, neural discriminability of CS+ and CS- responses of tufts increased significantly across learning (*Figure 3g*, blue; first-rewarded versus last-rewarded: $p<10^{-3}$, pre versus post: $p<10^{-4}$; Wilcoxon rank sum test). By contrast, neural discriminability of tuft responses in the repeated exposure mice decreased slightly with progressive exposure to the stimuli (*Figure 3g*, gray; Day 1 versus Final: p<0.01). Finally, we asked whether the ability to decode stimulus identity on a trial-by-trial basis increased after learning. To test this, we trained a support vector machine (SVM) to decode stimulus identity from tuft population activity (see Methods). We found that decoder performance increased significantly when comparing Pre and First sessions to Post and Last sessions (*Figure 3—figure supplement 2a*; sign-rank test, p=0.002), whereas decoder performance did not improve over time in the repeated exposure mice (*Figure 3—figure supplement 2b*; sign-rank test, p=0.22). Taken together, these results show that enhanced stimulus representations can emerge in apical tufts, but require reinforcement.

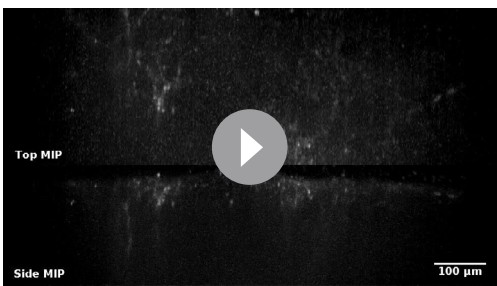

**Video 2.** Example SCAPE microscopy movie during behavioral session. Top, maximum intensity projection (MIP) across the dorsal-ventral dimension showing horizontal extent of dendritic activity. Bottom, MIP across the medial-lateral dimension showing vertical extent of dendritic activity. Playback speed is in real time. 'CS+' and 'CS-' denote times of stimulus onset. 300 × 1050 × 234 µm field of view.

https://elifesciences.org/articles/98349/figures#video2

The above analyses rely on the accurate measurement of calcium spikes from individual tufts. While two-photon microscopy acquires images with high resolution and speed, the imaging field is restricted to a single focal plane. This method can only measure calcium signals from a thin cross-section of the three-dimensionally complex apical structures. Indeed, many of the spatial components extracted from our two-photon data were comprised of dendritic branches that cross the imaging plane at different locations (*Figure 4—figure supplement 1a*), which makes it difficult to determine whether the segmentation software accurately extracted signals from one tuft or erroneously merged multiple tufts. For the same reasons, a single apical tuft could be falsely classified as two different tufts. Such errors could mislead our interpretation of selectivity in the population, especially given that a single apical tuft can exhibit non-homogenous branch-specific events (*Larkum et al., 2009*; *Cichon and Gan, 2015*; *Palmer et al., 2014*).

To confirm that our interpretation was not due to segmentation errors, we repeated the conditioning experiment using a high-speed volumetric imaging approach called SCAPE (*Bouchard et al., 2015*; *Hillman et al., 2018*), which allowed us to monitor calcium across entire apical tufts (*Video 2*). These three-dimensional datasets (300×1050 × 234 µm field of view) encompassed large portions of

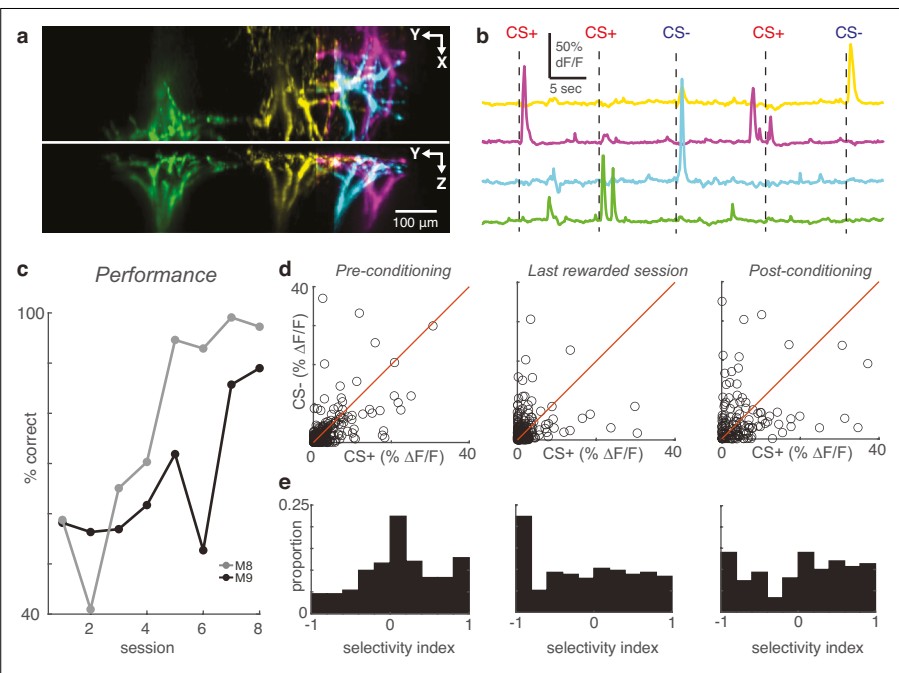

**Figure 4.** High-speed volumetric imaging of apical tufts confirms the emergence of enhanced selectivity after learning. (**a**) Top and side view of four example tufts segmented from volumetric SCAPE imaging. (**b**) Time courses of calcium activity from example tufts in (**a**) during five presentations of air puff stimuli (dashes). (**c**) Performance across all conditioning sessions of two mice that were imaged with SCAPE. (**d**) Across the indicated sessions, individual SCAPE-imaged tufts (circles) exhibit larger biases to CS+ or CS-. (**e**) Conditioning reshapes selectivity distribution from Normal to uniform.

The online version of this article includes the following figure supplement(s) for figure 4:

**Figure supplement 1.** Segmented tufts from two-photon and SCAPE microscopy.

the apical tree which included branches converging on their bifurcation points in layer 2, enabling us to identify whole apical trees unambiguously (*Figure 4a and b*; *Figure 4—figure supplement 1b*). CalmAn effectively demixed overlapping trees in these three-dimensional volumes. Using SCAPE microscopy, we imaged tuft activity of two additional mice conditioned with the same behavioral paradigm (*Figure 4c*). Comparison of tuft responses to the CS+ and CS- on the pre, last-rewarded, and post sessions (*Figure 4d*; 241 pre, 215 last-rewarded, 150 post tufts in 2 mice) revealed again that task learning induced significant increases in tuft selectivity (*Figure 4e*; pre versus last-rewarded: $p<10^{-5}$, pre versus post: $p<10^{-4}$, Wilcoxon rank sum test of |SI|). On average, the SI magnitudes were similar between tufts imaged using two-photon microscopy and SCAPE (mean ± s.e.m. |SI| for two-photon versus SCAPE; pre: 0.41±0.01 vs 0.40±0.02; last-rewarded: 0.54±0.02 vs 0.54±0.02; post: 0.51±0.02 vs 0.53±0.03). These data demonstrate that the effects in our two-photon dataset are not caused by

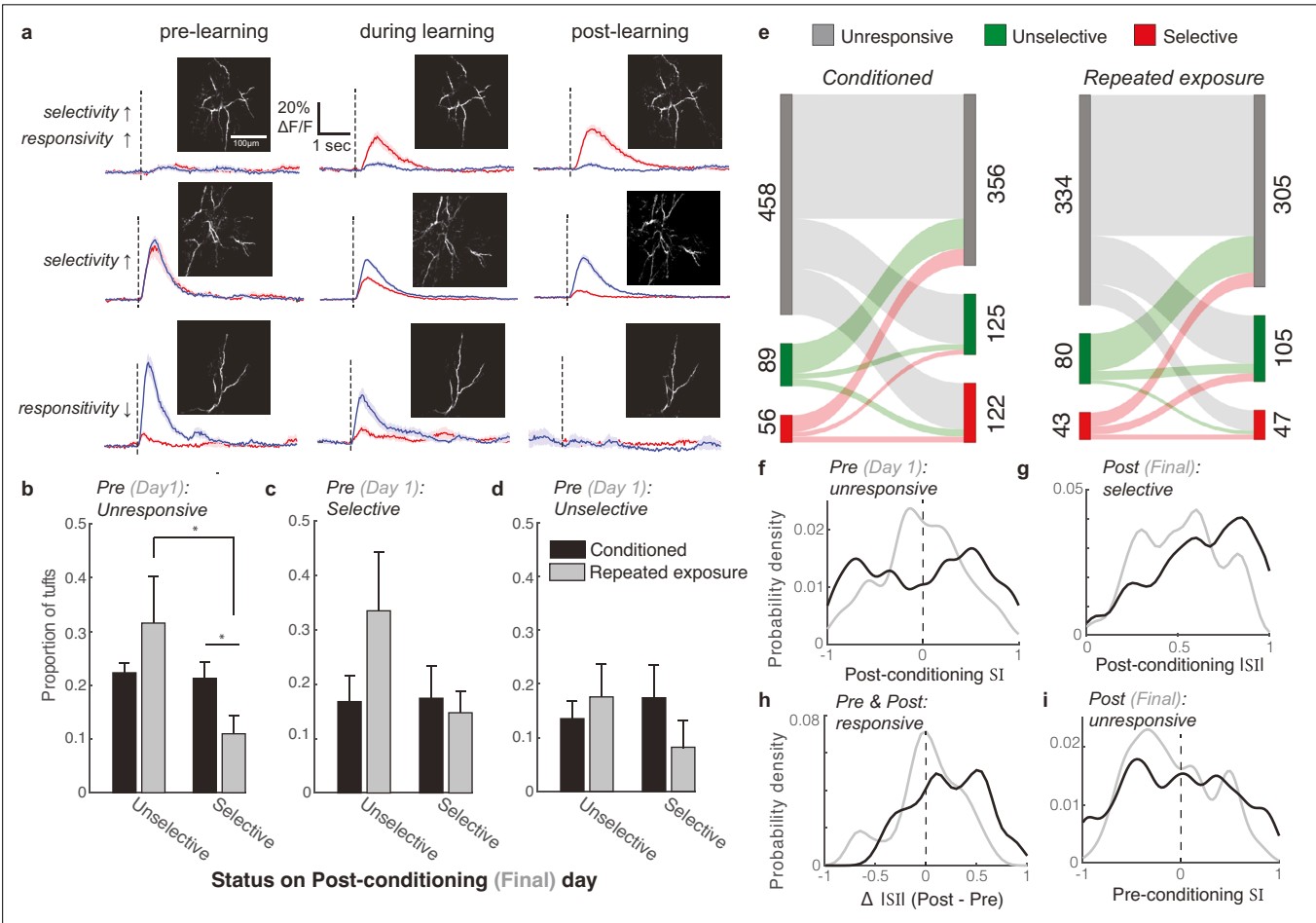

**Figure 5.** Longitudinal tracking reveals that reward enhances the selectivity of both initially unresponsive and responsive tufts. (**a**) Three example tufts that were longitudinally tracked across learning. Top row: An initially unresponsive tuft develops a robust response to the CS+ but not the CS- after learning. Middle row: A responsive but unselective tuft loses its robust CS+ response and becomes selective for the CS-. Bottom row: A CS- selective neuron becomes unresponsive to both stimuli. (**b**) Tufts that were unresponsive during the first session were longitudinally tracked to the last session. Plotted is the mean proportion of selective and unselective neurons across all animals in the conditioned (black bars) and repeated exposure (grey bars) groups. (**c,d**) Same analysis as in panel b for initially selective (**c**) and unselective (**d**) tufts. Two-sample t-test was used for comparisons between conditioned and repeated exposure groups. Paired t-test was used for comparisons within a group. * p<0.05. (**e**) Total tuft counts from first to last session within the three response categories for either conditioned (left) or repeated exposure (right) groups. (**f**) SI of responsive tufts on the last session that were initially unresponsive during the first session. Conditioned tufts have enhanced selectivity compared to repeated exposure. (**g**) Tufts that were selective on the last session are more selective if conditioned (black) rather than undergoing repeated exposure (grey). (**h**) Tufts that responded on both pre- and post-sessions tend to have higher selectivity if conditioned rather than undergoing repeated exposure. (**i**) SI of responsive tufts on the first session that later became unresponsive during the last session.

The online version of this article includes the following figure supplement(s) for figure 5:

**Figure supplement 1.** Calcium event rate of tufts that were either unresponsive or responsive to air puff stimuli.

errors in segmentation, but rather reflect changes at the level of individual dendritic tufts. Our results, based on two different imaging approaches, clearly demonstrate that reinforcement increases stimulus selectivity at the level of the entire apical tuft.

## Selective tufts emerge from both initially unresponsive and responsive populations

The striking effect of reinforcement learning on tuft response selectivity could develop in several ways. For example, initially unresponsive tufts could develop a robust response to either stimulus after learning (e.g. *Figure 5a*, top). Conceivably, tufts that were initially unselective in naive animals could also maintain their response to one stimulus while losing their response to the other (e.g. *Figure 5a*, middle). Either or both scenarios could lead to the increase in neurons that are selective for stimulus direction. To investigate which changes in individual tufts underlie population-wide improvements in stimulus selectivity, we longitudinally tracked the same set of tufts across all sessions and compared their selectivity in pre- and post-conditioning sessions for both conditioned and repeated exposure mice.

First, we categorized tufts that were unresponsive to either stimulus on the first imaging session, which accounted for the large majority of tufts (*Figure 5e*; conditioned: 458/603; repeated exposure: 334/457), and compared their response to the CS+ and CS- on the last session to determine if they became selective (*Figure 5b*, see Methods). Stimulus-unresponsive tufts, while on average less active than responsive ones (median calcium events per minute: 2.65 vs 3.66 for stimulus-unresponsive and responsive tufts, respectively; $p<10^{-40}$, Wilcoxon rank sum test; *Figure 5—figure supplement 1*), were not silent, with many undergoing tuft-wide calcium influx several times per minute. Silent tufts that are never active during the session may not have been detected in our imaging, but we were able to detect tufts that discharged as few as 3 voltage-gated calcium spikes over a 30-min behavioral session. Interestingly, in both the conditioned and repeated exposure mice, approximately 40% of initially unresponsive tufts developed a response to at least one stimulus by the last session, becoming either selective or unselective (*Figure 5b*). However, in conditioned animals, the proportion of initially unresponsive tufts that became selective was significantly larger than in repeated exposure mice (*Figure 5b*; p=0.04, 2-sample t-test comparing mice). Furthermore, while the proportion of selective and unselective tufts in this category was similar for conditioned animals, unselective tufts were more common in repeated exposure mice (*Figure 5b*; p=0.03, paired t-test).

Next, we analyzed tufts that were initially responsive and either selective (*Figure 5c*; conditioned: 56/603, repeated exposure: 43/457) or unselective (*Figure 5d*; conditioned: 89/603, repeated exposure: 80/457). In these smaller categories, we found no significant differences in the outcome of selectivity between the two groups of animals. Together, these results indicate that, while both stimulus exposure and reinforcement can alter tuft tuning, the presence of reward increases the likelihood that initially unresponsive tufts develop selectivity for either the CS+ or CS- (summarized in *Figure 5e*).

While a greater proportion of tufts from the conditioned animals were selective during the final session (20.2% vs 10.3% of tufts from conditioned and repeated exposure mice, respectively), we wondered whether conditioning also impacted the degree of selectivity. Note that some tufts had very small yet statistically different CS+ and CS- response amplitudes and were thus classified as selective despite a small SI. First, we compared the SI of initially unresponsive tufts on the final imaging session (*Figure 5f*). Supporting our results in *Figure 5b*, the SI distribution was shifted toward the tails in conditioned, but not repeated exposure mice, indicating that reward enhances selectivity for either the CS+ or CS- in this subset (|SI| conditioned versus repeated exposure: $p<10^{-5}$, Wilcoxon rank sum test, n=199 and 110 tufts, respectively).

Next, we compared the |SI| of all tufts that were categorized as selective during the last imaging session in conditioned and repeated exposure mice (*Figure 5g*). Interestingly, we found that even among selective tufts, the |SI| distribution in conditioned mice was significantly greater than in repeated exposure mice (p=0.006, Wilcoxon rank sum test, n=122 and 47 tufts, respectively), indicating that while selective tufts are present after both conditioning and repeated stimulus exposure, the magnitude of selectivity is stronger after conditioning.

We quantified the change in |SI| of all tufts that were responsive in both the first and last sessions by computing the difference between the two sessions (*Figure 5h*). Tufts in conditioned mice exhibited a greater increase in |SI| across sessions compared to repeated exposure mice (p=0.01, Wilcoxon rank

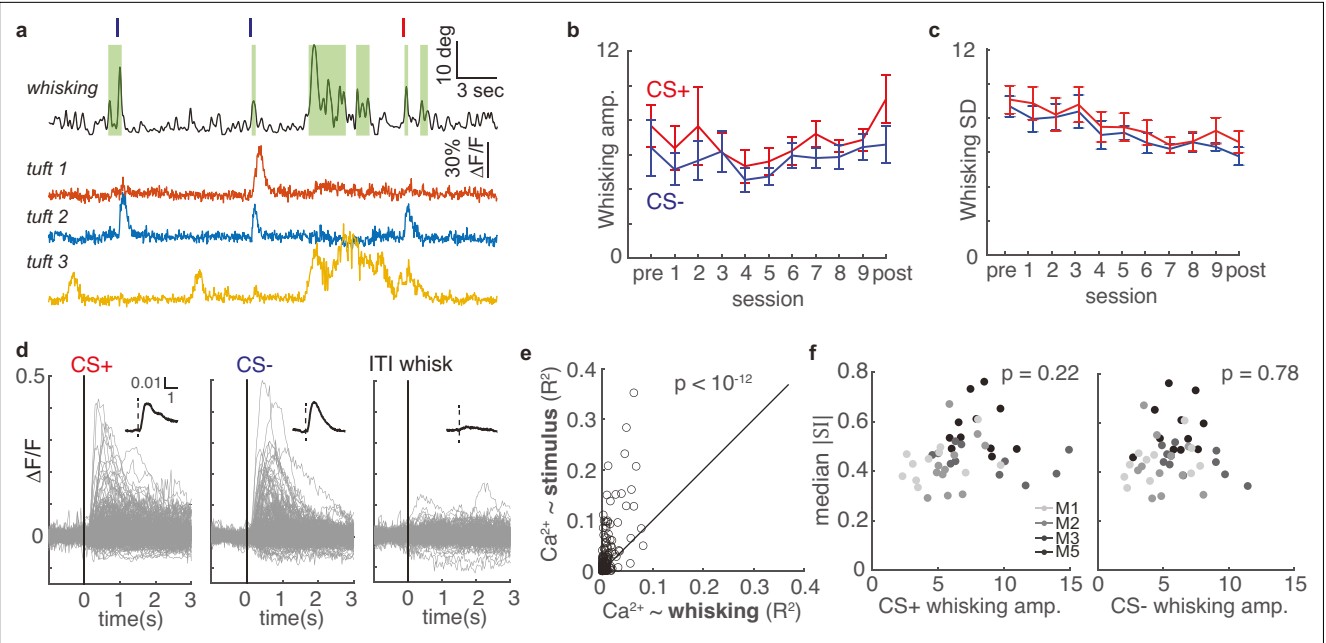

**Figure 6.** Whisking is only weakly correlated with tuft activity and cannot account for changes in selectivity during learning. (**a**) Whisking amplitude aligned to calcium activity of three example tufts in one session. Green shading indicates periods of whisking. Red and navy ticks indicate CS+ or CS- delivery, respectively. (**b**) Mean whisking response of five mice to CS+ (red) and CS- (navy) does not change across sessions during learning (mean ± s.e.m.). (**c**) Mean standard deviation of whisking decreases for both CS+ and CS- across learning, but CS+ and CS- do not differ. (**d**) Event-triggered averages of 322 tufts on the post-conditioning day (grey traces - individual tufts, black inset - population average) are responsive to stimuli but relatively unmodulated by whisking. (**e**) $R^2$ values for linear models predicting calcium from stimuli (y axis) are consistently greater than those predicting calcium from whisking (x axis). Each circle represents one of 322 tufts. (**f**) Magnitude of tuft selectivity does not correlate with mean whisking amplitude during CS+ (left) and CS- trials (right) on that session.

sum test, n=48 and 42 tufts, respectively), demonstrating that the magnitude of selectivity in initially responsive tufts increases after reinforcement learning.

The degree of selectivity of tufts that eventually became unresponsive on the last session was overall similar between the two groups (*Figure 5i*, |SI| conditioned versus repeated exposure: p=0.06, Wilcoxon rank sum test, n=97 and 81 tufts, respectively). However, tufts that became unresponsive were more likely to be initially highly selective in the conditioned group than in the repeated exposure group (19 tufts with initial |SI|>0.75 / 97 tufts ending as unresponsive in the conditioned group versus 3/81 in the repeated exposure group; p=0.0013, Z approximation to binomial). Therefore, learning can involve a loss of responsivity in a small subset of well-tuned tufts.

In summary, our longitudinal analyses revealed that reinforcement learning biases initially unresponsive tufts toward becoming selective and enhances the selectivity of tufts that are initially responsive.

## Neither movement nor behavioral choice account for enhanced selectivity

Several plausible factors could underlie the changes in selectivity we observed across learning. For instance, movements like whisking are correlated with layer 5 somatic action potentials (*Derdikman et al., 2006*; *de Kock and Sakmann, 2009*; *Rodgers et al., 2021*) and might have impacted calcium activity in the apical tuft. To investigate whether whisking could account for the changes in tuft selectivity, we imaged the whiskers with a high-speed camera and computed whisking amplitude (see Methods) while mice underwent conditioning and two-photon imaging (*Figure 6*). First, we considered whether animals changed their whisker movements in response to conditioned stimuli over the course of learning. We computed the peak of the mean stimulus-aligned whisking amplitude for the CS+ and CS- (*Figure 1c*, left; *Figure 6b*) for each session in five mice. Although conditioning alters licking behavior (*Figure 1c and e*), the magnitudes of whisker movements following both stimuli were stable across sessions (*Figure 6b*; CS+: p=0.44; CS-: p=0.45; linear regression). We also computed

the standard deviation (SD) of stimulus-evoked whisker amplitude across trials for all sessions (*Figure 6c*). While the whisking amplitude became slightly more reliable (decreased SD) across sessions (p<10⁻⁴), the change in reliability across sessions was similar for CS+ and CS- (p=0.53). Therefore, whisking is similar on both trial types throughout learning.

We next examined whether whisking was correlated with tuft calcium activity by comparing stimulus-triggered averages and intertrial interval (ITI) whisk-triggered averages of all tufts during post-conditioning. Whisking amplitude was similar between spontaneous ITI whisking bouts and evoked whisking responses to stimuli (n=115 and 617 events, respectively; p=0.53, Wilcoxon rank sum test). In contrast to air puff stimuli, ITI whisking bouts were not associated with a robust calcium response (*Figure 6d*).

To quantify the relationship of whisking and sensory stimuli to tuft calcium spikes, we performed a linear regression analysis (see Methods) on 322 tufts using calcium influx as the response variable and either stimulus or whisking amplitude as a single predictor variable (*Figure 6e*). Air puff stimuli more reliably predicted calcium influx than whisking amplitude for each of virtually all tufts (p<10⁻¹², sign rank test). These results are consistent with other studies that found either only weak or no correlation between whisking and L5 tuft calcium spikes in S1 (*Lacefield et al., 2019*; *Xu et al., 2012*; *Takahashi et al., 2020*). Furthermore, we found no relationship between the whisking response and the median SI magnitude on a given session (*Figure 6f*, whisking to CS+ p = 0.22, CS- p=0.78). Therefore, changes in whisker movement cannot account for the changes in selectivity during learning that we observed.

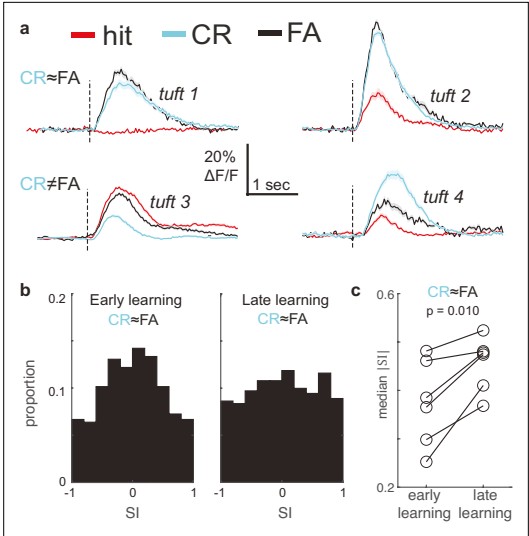

**Figure 7.** Behavioral responses do not account for enhancement of stimulus selectivity during learning. (**a**) Mean stimulus responses of four tufts during hit (red), CR (cyan), and FA (black) trials. Top row: Example tufts whose responses are not behaviorally modulated (CR is similar to FA). Bottom row: Example tufts with behaviorally modulated responses (CR and FA differ). (**b**) Selectivity index (SI) distribution changes from early (left) and late learning sessions (right) even when tufts with behaviorally modulated responses (CR≠FA) are excluded. (**c**) Median SI magnitude of tufts in each of six animals (from panel **b**) increases from early to late learning sessions.

The online version of this article includes the following figure supplement(s) for figure 7:

**Figure supplement 1.** Licking cannot account for changes in selectivity during learning.

Finally, the possibility remains that other task-related signals relaying information about reward expectation and behavioral choice could impact apical tuft activity and drive increases in selectivity. To test this, we compared tuft responses to the CS- in false alarm trials (FA; mouse incorrectly licked for reward) and correct rejection trials (CR; mouse correctly withheld licks) to determine if their activity was modulated by behavioral choice. Notice that these two trial types have the same sensory input but involve different choices. (The corresponding analysis for CS+ trials is not technically possible for lack of sufficient Miss trials after the first conditioning day, an issue also observed in *Poort et al., 2015*. A future experiment in which the stimulus strengths are substantially reduced would drastically increase the error rates, enabling a comparison between Hit and Miss trials.) Tufts were classified as behaviorally modulated if the FA response was significantly different from the CR response, and were not behaviorally modulated if CR and FA responses were statistically indistinguishable (e.g. *Figure 7a*). Behaviorally modulated tufts accounted for only ~10% of the total tuft population in both early and late learning (50/395 in early; 35/406 in late learning).

To test whether these behaviorally modulated tufts contributed to increased selectivity during learning, we excluded them and compared selectivity of the remaining behaviorally-insensitive tufts. We found that selectivity increased significantly from early to late learning (*Figure 7b and c*; median |SI| of 345 tufts early versus 371 tufts late learning: 0.38 vs 0.47, p=0.02, Wilcoxon rank sum test), similar to our previous analysis of the entire population. Licking, like whisking, was a relatively poor

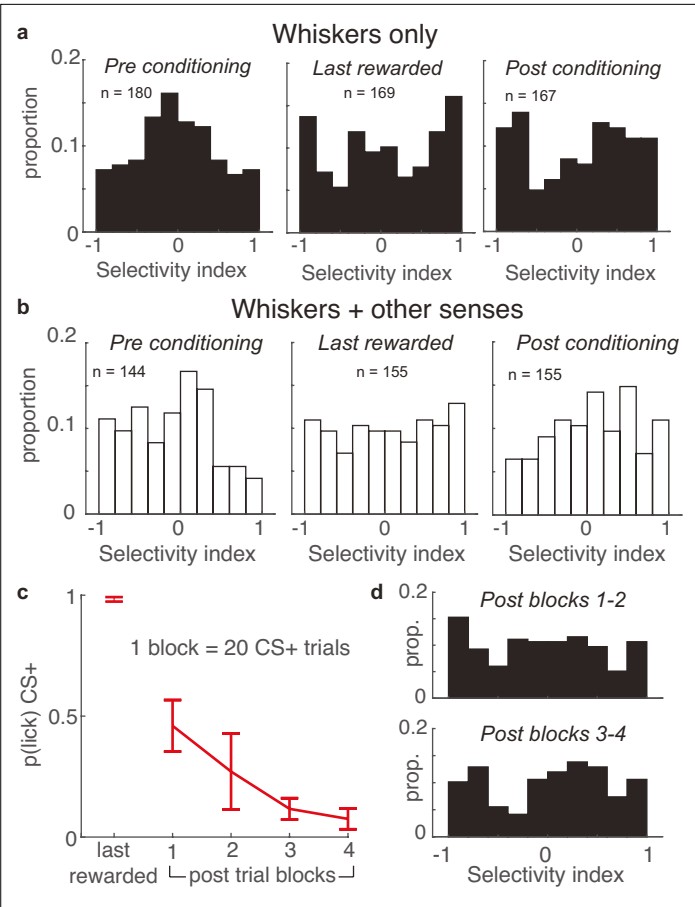

**Figure 8.** Apical tufts in barrel cortex of mice performing the task exclusively with their whiskers undergo long-lasting changes in selectivity. (**a**) SI histograms of mice performing the task exclusively with their whiskers exhibit increased selectivity across pre-conditioning, last-rewarded, and post-conditioning sessions. (**b**) Relative to pre-conditioning, mice using their whiskers and other sensory cues to perform the task have increased selectivity during the last rewarded session, but not the post-conditioning session. (**c**) The probability of anticipatory licks in response to the CS+ extinguishes across post-conditioning blocks (of 20 trials each). (**d**) Tuft selectively remains uniformly distributed during post-conditioning trial blocks 1–2 (top) while licking is extinguishing, and blocks 3–4 (bottom) in which licking is extinguished.

predictor of tuft calcium influx (*Figure 7—figure supplement 1a and b*). Because some behaviorally modulated tufts may not have been statistically detectable, we used multivariate linear regression to disentangle stimulus responses from licking and whisking, which may have been confounded with choice. Median coefficients for licking and whisking were on average 3.3 times smaller than median stimulus coefficients for the first rewarded, last rewarded, and post sessions (all $p<10^{-6}$, Wilcoxon rank sum test). Even after we factored out possible effects of movements, CS+ and CS- coefficients were enhanced by learning but not repeated exposure (*Figure 7—figure supplement 1c and d*), consistent with our other analyses. Together, these results demonstrate that enhanced selectivity during learning cannot be explained by non-sensory signals related to the animals' behavior.

## Enhanced selectivity in barrel cortex is long-lasting when mice exclusively use whiskers

Mice could conceivably exploit other sensory cues to learn and perform the task, such as auditory cues from the air nozzles or non-whisker tactile cues from air current eddies contacting the fur or skin. To determine which mice exclusively used their whiskers to distinguish the CS+ and CS-, we trimmed all whiskers after the post-conditioning session and assessed performance in five mice (*Figure 8*). Performance in each of the five mice decreased after whisker trimming, indicating that each used some

whisker information. Three mice performed the task exclusively with their whiskers, falling to chance levels after the whisker trim ('whiskers only'). Two other mice still performed the task above chance after the whisker trim, indicating that they were not exclusively using their whiskers and exploited information from multiple sensory streams ('whiskers + other senses').

We examined whether these two different behavioral strategies impacted tuft selectivity. Both the 'whiskers only' and 'whiskers + other senses' groups exhibited enhanced tuft selectivity in the last-rewarded session relative to pre-conditioning. This effect was more pronounced in the 'whiskers only' mice (*Figure 8a and b*, left and middle; whiskers only: median |SI| of 180 pre tufts versus 169 last-rewarded tufts: 0.36 vs 0.59, p<10$^{-3}$; 'whiskers + other senses': median |SI| of 144 pre tufts versus 155 last-rewarded tufts: 0.39 vs 0.50, p=0.01). Surprisingly, enhanced selectivity persisted during the post-conditioning session for the 'whiskers only' group but not the 'whiskers + other senses' group (*Figure 8a and b* right panels; whiskers only: median |SI| of pre versus 167 tufts post: 0.36 vs 0.58; p<10$^{-3}$; whiskers + other senses: median |SI| of 155 pre versus post tufts: 0.39 vs 0.42; p=0.45). Therefore, tuft selectivity in barrel cortex is enhanced regardless of behavioral strategy, but outlasts conditioning only when mice rely solely on their whiskers to perform the task.

We further examined this persistence of enhanced tuft selectivity as experienced mice stopped performing the task. While the entire post-conditioning session was unrewarded, mice initially expected rewards and licked for many CS+ trials in the first half of the session. By the second half of the session, the probability of a lick occurring during the CS+ extinguished, approaching zero (*Figure 8c*). We compared the selectivity of tufts during the first and second halves of the post-conditioning sessions of mice that exclusively used their whiskers and found no difference in the two distributions (*Figure 8d*, p=0.94, Wilcoxon rank sum test of |SI|), demonstrating that selectivity of the population remained stable throughout the session. Taken together, these results demonstrate that enhanced stimulus selectivity of apical tuft dendrites after reinforcement learning is long lasting, persisting even after mice cease performing the task and expecting reward.

## Discussion

Our study is the first to investigate how learning a discrimination task alters apical tuft activity. Using both novel volumetric whole-tuft imaging and conventional planar microscopy, we discovered that L5 apical tufts acquire enhanced representations of multiple stimuli during learning. Rather than simply retuning tufts toward the rewarded stimulus, learning enhanced selectivity for both stimuli, suggesting that tufts are aligning themselves to the behaviorally relevant stimulus dimensions. This effect is particularly striking given that, by design, the two stimuli are in orthogonal not opposing directions. These enhanced sensory representations persist even after mice cease performing the task. In contrast, representations are slightly degraded by mere repeated exposure to stimuli outside of a task. Consistent with previous studies (*Lacefield et al., 2019*; *Xu et al., 2012*), we found that movement in and of itself has little direct impact on tuft spikes, indicating that increased selectivity of apicals reflects alterations in sensory coding as animals learn. This sensitization of tufts to behaviorally relevant sensory dimensions may be a general feature of all sensory cortical areas.

Tuft spikes enhance plasticity of synaptic inputs that occur over behavioral (seconds-long) timescales (*Roelfsema and Holtmaat, 2018*; *Bittner et al., 2017*). These new behaviorally relevant tuft representations may therefore prime subsequent plasticity of synapses across the entire pyramidal neuron. Additionally, tuft events potently modulate somatic burst firing and enhance how somata respond to their basal inputs (*Larkum et al., 2009*; *Larkum, 2013*). As learning and plasticity increase apical selectivity for a behaviorally relevant axis, tuft events will unavoidably amplify somatic burst output along the same axis. This could enable action potential output of L5 cells in primary sensory cortex to directly drive behavioral responses via projections to movement related areas, such as the corticostriatal, corticopontine, and corticotrigeminal pathways. Thus, tuft spikes have the potential to modify somatic output, both in the present and in the future.

An open question is whether enhanced stimulus representations in apical tufts are required for learning this task. One way to address this question would be to silence tuft activity during and after learning by optogenetically activating NDNF-positive interneurons in layer 1 (*Abs et al., 2018*). This approach is not ideal as NDNF interneurons also inhibit other cells such as layer 2/3 pyramidal cells, PV interneurons (*Cohen-Kashi Malina et al., 2021*), and possibly the axons of layer 5 pyramidal cells, which are known to densely innervate layer 1. Because this manipulation is not specific to layer 5

apicals, the results would be difficult to interpret. Focal illumination of inhibitory opsins in tufts has also been used to assess tuft function (*Ranganathan et al., 2018*), but balancing tuft against soma silencing remains challenging and complicates interpretation. Better tools for selective targeting of apicals would be extremely useful for addressing such issues.

## Enhanced representation of behaviorally relevant stimuli

Enhancing the representation of relevant stimulus dimensions rather than a singularly important stimulus, such as a rewarded event, has multiple benefits for behavior. In our paradigm, both the CS+ and CS- are predictive of whether or not a reward will occur in the future. Explicitly encoding both stimuli could allow sensory cortical areas to directly elicit actions. In the context of this task, CS+ preferring tufts in barrel cortex may trigger anticipatory licking while CS- preferring tufts could suppress licking. L5 cells in sensory cortex via their output to striatum, pons, brain stem, and spinal cord would thereby be able to directly and rapidly drive action without further cortical processing, such as by frontal areas including motor cortex (*Takahashi et al., 2020*; *Park et al., 2020*). Such rapid sensory-motor transformations by primary sensory areas may be critical for natural time-constrained behavior.

Furthermore, learning produced a representation in which the degree of selectivity for the two stimuli was continuous and uniformly distributed. Exclusively CS+ or CS- selective apicals never dominated the population. Continuous degrees of selectivity across the population, rather than discrete representations, may allow the system to be more robust to the variability caused by active movements that alter sensory input. A continuous distribution may also facilitate future adjustments of neural representations as subjects continue to learn a task or encounter new tasks. The uniformity we observed may reflect that neurons are high-dimensional, being sensitive to mixtures of variables (*Rodgers et al., 2021*; *Rigotti et al., 2013*; *Stringer et al., 2019*; *Kim et al., 2020*), only one of which might be altered here by learning. The uniform distribution of selectivity corresponds to a full range of pessimism to optimism concerning stimulus predictions of upcoming rewards. Recent work shows that behavioral performance benefits from reinforcement learning that incorporates the distribution of reward probabilities rather than just the average expected reward value (*Dabney et al., 2020*). L5 corticostriatal synapses could theoretically afford a plastic substrate for acquiring the necessary distribution of reward probabilities.

Surprisingly, past studies in which mice were trained to associate one or more stimuli with a reward typically show that cortical representations are stronger for the rewarded stimulus (*Poort et al., 2015*; *Henschke et al., 2020*; *Goltstein et al., 2013*). In contrast to these studies of layer 2/3 somatic activity, our experiments revealed that the overall tuft calcium response to the CS+ and CS- at the population level did not change significantly after animals learned the task (*Figure 2*). Instead, representations for both stimuli were enhanced by individual tufts developing selectivity for either the CS+ or the CS- (*Figure 3*). This divergence in phenomena may result from several important differences between our work and the aforementioned studies.

First, enhanced selectivity for both rewarded and unrewarded stimuli could be a phenomenon that is unique to the apical dendritic tufts. In addition to local inputs, the apical tufts of pyramidal cells in S1 receive long-range top-down input from several sources, including motor cortex (*Xu et al., 2012*; *Petreanu et al., 2012*), secondary somatosensory cortex (*Cauller et al., 1998*), and secondary thalamus (*Zhang and Bruno, 2019*; *Rubio-Garrido et al., 2009*; *Wimmer et al., 2010*). Frontal areas, such as prefrontal cortex, indeed have enhanced representations of the CS+ and CS- after learning (*Wang et al., 2020*). In contrast, input to the somata is dominated by the local cortical area and primary thalamus (*Feldmeyer, 2012*; *Constantinople and Bruno, 2013*). While somato-dendritic coupling can be strong in L5 neurons (*Beaulieu-Laroche et al., 2019*), it is asymmetric; at least 40% of somatic transients attenuate in a distance-dependent manner along the apical trunk and distal tufts (*Francioni et al., 2019*). The non-overlapping anatomical inputs and asymmetric coupling together could produce different learning-related effects on apical tuft and somatic stimulus representations.

Second, learning-related changes may manifest differently in layer 2/3, the usual focus of previous studies (*Poort et al., 2015*; *Henschke et al., 2020*), and layer 5 pyramidal cells, the tufts of which we studied. With the exception of a small population of corticostriatal cells, most excitatory cells in layer 2/3 project to other cortical areas to affect further cortical processing (*Petersen and Crochet, 2013*; *Yamashita et al., 2018*). In contrast, many L5 cells project to subcortical structures including the thalamus, superior colliculus, and brainstem, which may directly trigger behavioral responses (*Llinás*

*et al., 1998*; *Krauzlis et al., 2013*; *Parvizi and Damasio, 2001*). In discrimination paradigms, both stimuli are relevant to behavior. In our task, the CS+ prompted licking to obtain a reward, and the CS- suppressed licking that would have no benefit. Thus, an enhanced representation of both stimuli in layer 5 would be advantageous for animals to perform the task efficiently. Recently, it was shown that apical dendrite activation of subcortical-targeting pyramidal tract L5 cells, but not intratelencephalic L5 cells that are more like L2/3 cells in their connectivity, determines the detection of tactile stimuli (*Takahashi et al., 2020*). The Rbp4-Cre mice we used in this study labels a heterogenous population of layer 5 pyramidal cells, comprising both pyramidal tract and intratelencephalic neurons. In the future, it would be interesting to examine whether learning has different effects on the sensory representations of these two populations. Moreover, direct comparisons of the layers would be particularly informative.

Finally, it is possible that learning-related changes in sensory representations manifest differently between a somatosensory modality and a visual modality, the latter being the focus of previous studies. To our knowledge, we are the first to show changes of sensory representations in somatosensory cortex within a discrimination paradigm. Facial stimuli are generally more salient than arbitrary visual stimuli for any species, and mice are known to be highly responsive to their tactile sense more so than vision (*Petty and Bruno, 2024*). This strong salience of whisker-mediated touch may make it particularly advantageous to develop sensory representations of a larger variety of relevant tactile stimuli, in this case, both the CS+ and CS-.

## Candidate plasticity mechanisms

Enhanced selectivity could be due to changes in local synaptic connectivity, long-range inputs, or both. Learning may strengthen and weaken synapses onto barrel cortex neurons from ascending thalamocortical input or from neighboring cells. Such local plasticity could enhance CS+ or CS- responsiveness. Alternatively or additionally, other cortical regions encoding task context could via long-range inputs reconfigure barrel cortex to respond more strongly to these stimuli. The present results do not completely distinguish between these two scenarios because long-range inputs may still encode the context while the mouse is in the behavioral apparatus. However, we found that enhanced representations persist after mice are no longer engaged in the task and receiving rewards. This result suggests that enhanced representations may be a product of local plasticity in sensory cortex that alters receptive fields.

Even in the absence of reward, repeated exposure to stimuli can drive plasticity in sensory cortex and alter response tuning. For instance, repeated exposure to oriented gratings can alter the orientation tuning of cells in primary visual cortex (*Yao and Dan, 2001*; *Dragoi et al., 2002*; *Dragoi et al., 2000*), and overstimulation of whiskers induces plasticity at dendritic spines and alters whisker representations in somatosensory cortex (*Zhang et al., 2015*; *Feldman and Brecht, 2005*). Our results demonstrate that at the population level enhanced representations developed only when stimuli were behaviorally relevant. Our longitudinal analysis revealed that while the response dynamics of some tufts changed after repeated stimuli presentations, overall selectivity of the population did not increase when rewards were omitted (*Figures 3 and 5*). This raises the question: What are the mechanisms that drive enhanced selectivity under rewarded conditions? In one possible scenario, reward delivery causes the release of neuromodulators that augment the activity of apical tufts. Cortical layer 1 is innervated by cholinergic afferents from the nucleus basalis (*Mechawar et al., 2000*) and adrenergic afferents from the locus coeruleus (*Freedman et al., 1975*), the main source of acetylcholine and norepinephrine, respectively. Salient events such as reward and arousal lead to the release of these neuromodulators in cortex (*Chubykin et al., 2013*; *Thiele and Bellgrove, 2018*), which could increase the excitability of apical dendrites by recruiting disinhibitory circuits or directly influencing dendritic currents (*Labarrera et al., 2018*; *Brombas et al., 2014*; *Chubykin et al., 2013*; *Hangya et al., 2015*). In this model, the release of reward-driven neuromodulators promotes plasticity and an enhanced representation of temporally aligned sensory inputs. This phenomenon was demonstrated in auditory cortex, where tones paired with stimulation of the nucleus basalis shifted the tuning of neurons toward the frequency of the paired stimulus (*Froemke et al., 2007*).

Why are representations of the CS- equally enhanced when there is no associated reward? One explanation is that, as mice learn that the CS- indicates absence of reward, the CS- effectively signals punishment and acquires negative value. Acetylcholine is released in response to aversive stimuli, and

can activate disinhibitory microcircuits that reduce inhibition onto pyramidal cells and may be essential for learning (*Letzkus et al., 2011*; *Gasselin et al., 2021*). Thus, it is possible that both the CS+ and CS- representations are enhanced by neuromodulatory mechanisms tied to reward and punishment, respectively. An open question is whether the outcome is due to reinforcement learning or the behavioral state brought on by the reinforcers rather than their valence. Sensory cortical plasticity may not be tied to reinforcer valence. Our paradigm creates an environment where mice benefit from being attentive and engaged in order to maximize reward while minimizing effort. Previous work has shown that active engagement in a visual discrimination task was associated with significantly higher selectivity in layer 2/3 cells in visual cortex (*Poort et al., 2015*). Task engagement may lead to a sustained increase in neuromodulator release throughout the conditioning session, priming the apical dendrites for plasticity and the development of selective responses for task-relevant stimuli as they learn.

What determines whether a particular tuft eventually becomes selective for the CS+ or CS-? Our longitudinal analysis revealed that many tufts that were initially unresponsive to either stimulus developed a highly selective response to either the CS+ or the CS- (*Figure 5*). In these tufts, stimulus preference after learning might be seeded by initially weak, directionally selective inputs on to the neuron that already exist prior to conditioning and that are potentiated by the learning process. We also found tufts that initially exhibited robust responses to both stimuli and either lost or significantly reduced their response to one stimulus after learning. The reduction of an apical response to a particular stimulus could be driven by local disynaptic inhibition between L5 pyramidal cells mediated by the apical-targeting Martinotti cells (*Berger et al., 2010*; *Kapfer et al., 2007*; *Naka and Adesnik, 2016*). Through this mechanism, L5 neurons that are selective for a particular stimulus could inhibit responses to that stimulus in neighboring L5 apical tufts. Experiments that assess the tuning of excitatory and inhibitory inputs onto apical dendrites as a function of learning could test such mechanisms.

In addition to demonstrating increased tuft selectivity with learning, we replicated a surprising phenomenon in a previous instrumental behavior in which a population of apical tufts exhibit activity around the time of reward (*Lacefield et al., 2019*). This reward-related activity was observed in four out of the seven conditioned animals only during CS+ trials and was most prominent during intermediate conditioning sessions, when most animals were still performing at chance levels, and disappeared completely by the final conditioning session (*Figure 2—figure supplement 1*). Other than this transient effect, unconditioned stimuli did not appear to elicit calcium responses, consistent with our previous findings (*Lacefield et al., 2019*). The disappearance of this reward-related peak might be attributable to the reward becoming predictable in later stages of learning. In previous classical conditioning experiments, dopaminergic cells exhibit responses to rewards early in learning due to the novelty of an unexpected stimulus. These responses are lost after extended training, as animals learn the association between the CS and reward (*Ljungberg et al., 1992*; *Pan et al., 2005*). While dopaminergic terminals are sparse in primary sensory areas, they are not entirely absent, nor are dopaminergic receptors. Furthermore, the excitability of the apical tuft is sensitive to noradrenaline (*Labarrera et al., 2018*). Interestingly, noradrenergic neurons in the locus coeruleus exhibit a similar phenomenon to dopaminergic neurons, where responses shift from temporal alignment with the reward to a predictive conditioned stimulus after learning (*Bouret and Sara, 2004*). Such mechanisms could explain why reward-related activity is restricted to early-to-intermediate learning in our paradigm.

## Global versus local dendritic spikes

Apical dendrites exhibit not only global spikes that elicit calcium influx across the entire tuft, which we exclusively analyzed here, but also local events known as NMDA spikes, which typically engage short (<30 μm) segments of individual dendritic branches (*Larkum et al., 2009*; *Xu et al., 2012*; *Palmer et al., 2014*). These local, NMDA receptor-dependent events can promote prolonged plasticity within individual dendritic branches in the absence of backpropagating actions potentials, a feature that is unique to the apical dendrites (*Sandler et al., 2016*). In motor cortex, branch-specific NMDA spikes are crucial for establishing the long-lasting plasticity necessary for learning (*Cichon and Gan, 2015*), and depolarization provided by multiple local NMDA spikes is thought to be essential for the generation of a global calcium spike triggered by distal synaptic inputs (*Larkum et al., 2009*). We focused this study on global tuft-wide calcium events, rather than local events. Local events are more difficult to unambiguously identify in planar imaging (*Sheffield and Dombeck, 2015*), and their existence in

vivo is still an open question for L5 apicals in barrel cortex (*Xu et al., 2012*; *Palmer et al., 2014*). Nonetheless, they may play important roles in plasticity processes that eventually lead to the emergence of global tuft spike selectivity for stimuli. Volumetric microscopy studies, the feasibility of which we showed here, are needed to further investigate the existence of local events in such behaviors as well as examine possible relationships between local and global tuft events during reinforcement learning. However, it would be essential to verify that seemingly spatially overlapping local and global events derive from the same dendritic tree, which requires greater resolution than was practical for the present study.

To analyze activity of individual tufts, we segmented these structures based on spatiotemporal covariance (*Giovannucci et al., 2019*). This method does not discount the possibility of errors where one tuft is split erroneously into two trees, or where two highly correlated tufts are merged. With this in mind, we used volumetric imaging SCAPE microscopy, which allowed us to visualize the apicals in three dimensions and unambiguously screen for such artifacts. The results from SCAPE are quantitatively similar to those from two-photon microscopy, and confirm that our observation of enhanced selectivity with learning is not an artifact of planar imaging.

## Stability of learned tuft representations

In contrast to previous studies of discrimination learning (*Poort et al., 2015*; *Liu et al., 2020*; *Henschke et al., 2020*), we included an unrewarded post-conditioning session to examine whether learning-related effects persisted through extinction. Our results show that post-conditioning selectivity of the apical population remains significantly higher than pre-conditioning, even after animals stop licking in response to the CS+ (*Figure 8*). Interestingly, the effects of learning are much more pronounced in animals that relied exclusively on their whiskers to perform the task. In animals that apparently used other sensory modalities, we observed a modest increase from the pre to last-rewarded session, which seemed to be largely absent by the post-conditioning session. Considering that these animals were additionally exploiting other sensory areas to perform, selectivity may have been more widely distributed and thus diluted in barrel cortex, diminishing the effect and its stability. How long selectivity persists in the neuronal population after conditioning and which factors influence stability are interesting open questions for future study.

## Conclusion

In summary, we have shown for the first time that reinforcement learning enhances representations along behaviorally relevant dimensions in apical tufts. Our results suggest that dendritic calcium spikes are an important cellular mechanism underlying the changes in sensory encoding that occur with learning, and provide an avenue for further investigation of cellular and circuit mechanisms underlying plasticity induced by perceptual experience and reinforcement. This cellular compartment may be key to understanding pathology in some cognitive, memory, and learning disorders.

## Methods

**Key resources table**

| Reagent type (species) or resource | Designation | Source or reference | Identifiers | Additional information |
|---|---|---|---|---|
| Genetic reagent (*Mus musculus*) | Rbp4-Cre | GENSAT | Rbp4-Cre_KL100 | |
| Genetic reagent | AAV1-CAG-flex-GCaMP6f | UPenn Vector Core | | |
| Software, algorithm | MATLAB | MathWorks, Inc. | v2022b | |
| Software, algorithm | CalmAn | https://github.com/flatironinstitute/CalmAn | v1.8.3 | RRID:SCR_021533 |
| Software, algorithm | Arduino IDE | https://www.arduino.cc/ | v1.8 | |

All experiments complied with the NIH Guide for the Care and Use of Laboratory Animals and were approved by the Institutional Animal Care and Use Committee of Columbia University (protocol AC-AABP0555). Sixteen C57BL/6 mice ranging in age from 77 to 316 days old (mean of 123 days at

the time of imaging) were used in these experiments. Six were male, and 10 female. Our results were observed in both male and female individuals, and no sex difference was detected.

## Surgery

Animals were administered dexamethasone (1 mg/kg) via intramuscular injection 1–4 hr prior to surgery to reduce edema. Anesthesia was induced with 3% isoflurane in oxygen and maintained at 1%. Mice were head-fixed in a stereotax, and a subcutaneous injection of bupivacaine (0.5%, 0.1 mL) was administered under the scalp. Buprenorphine (0.05 mg/kg) was injected subcutaneously on the back. The scalp was cut, and the skull was covered with a thin layer of Vetbond. A circular craniotomy (3 mm diameter) centered at 1.5 mm posterior and 3.5 mm lateral to bregma was made using a dental drill. The dura was kept moist using artificial cerebrospinal fluid.

For both two-photon and SCAPE microscopy, Rbp4-Cre_KL100 mice were injected with 100 nL of virus (initial titer ~2 × 10¹³ cfu/mL, diluted 1:4 in artificial cerebrospinal fluid) encoding GCaMP6f in a Cre recombinase-specific manner (AAV1-CAG-flex-GCaMP6f, UPenn Vector Core). The virus was injected in layer 5B of the barrel cortex (1.0 mm deep to the pia) using a pulled pipette (20–30 μm ID) fastened on a Nanoject III, which was mounted on a manipulator angled at ~30° from vertical. The virus was delivered via four injections of 100 nL each, spaced at least 400 μm apart. The depth was chosen to maximize labeling of thick-tufted pyramidal neurons. In pilot experiments, we found that placing injections 1.0 mm deep resulted primarily in thick-tufted labeling whereas at more superficial depths (e.g. 0.8 mm deep) we obtained mainly thin-tufted tufts, consistent with *Oberlaender et al., 2012*. The dura was then removed, and a thin cover glass was implanted and sealed using super-glue. A custom metal head plate was implanted on the skull using dental cement. Twenty-four hours after surgery, carprofen (5 mg/kg) was administered subcutaneously. Imaging and behavioral training commenced 3 weeks after surgery. Animals were only excluded from the study if the cranial window quality was poor, which precluded imaging.

## Behavior

Animals in both rewarded 'conditioning' and unrewarded 'repeated exposure' groups were water restricted for 2 days prior to starting imaging and habituated to head fixation for ~10 min on each of these 2 days. They were subsequently given ~1 mL of water per day for 9 days either by pairing water rewards with a specific stimulus (conditioning group), or in their cage following the imaging session (repeated exposure group). Mice were head restrained in a custom-made behavioral apparatus by positioning the body in a 3D-printed chamber and fastening the head plate to metal posts flanking the chamber. Air puff stimuli (10 psi measured before a control solenoid, 100ms) were delivered from two nozzles (cut P200 pipette tips) positioned toward the distal tips of the whiskers, in either the rostrocaudal or ventrodorsal direction. Nozzles were oriented to prevent air jets from stimulating other parts of the face. One of these directions (CS+) was paired with a water reward (10 μL), delivered through a lick port 0.5 s after the stimulus onset. The particular direction (rostrocaudal vs ventrodorsal) used as the CS+ was randomized and counterbalanced across mice. Approximately 180 stimuli were presented over the course of a 30 min imaging session (8–12 s intertrial interval). The probability of CS+ or CS- delivery was 50%. In preliminary experiments, we found that an auditory mask helped prevent mice from exploiting auditory cues to discriminate the two stimuli: a third air nozzle was positioned close to the mouse and was active throughout the session.

During the first session (pre-conditioning), stimuli were delivered in the absence of reward to assess neural and behavioral responses in naïve animals. In the following 7–9 days, the CS+ was paired with reward. Licks for rewards were detected with a capacitance-based touch sensor (Sparkfun). A trial response was registered when one or more licks were elicited within a 0.5 s response window following the stimulus and before reward delivery. To determine whether behavioral performance was above chance, we computed 95% confidence intervals using the 'binofit' function in MATLAB. During the final session (post-conditioning), stimuli were delivered in the absence of reward. Animals in the unrewarded group received the same two stimuli across 9 days without reward pairing. Behavioral experiments were performed with the Arduino-based OpenMaze open-source behavioral system, whose designs are fully described at https://www.openmaze.org/. Whisking was monitored at 125 fps with a camera (Sony PS3eye) and automatically tracked using published software (*Clack et al., 2012*).

## Intrinsic signal optical imaging and two-photon imaging

Intrinsic signal optical imaging and two-photon imaging were performed on a Sutter movable objective microscope. The locations of whisker barrels in S1 were identified using intrinsic signal optical imaging. Single whiskers in isoflurane-anesthetized mice were stimulated at 5 Hz using a piezoelectric bimorph while recording the reflectance of 700 nm long-pass incandescent light with a Rolera CCD camera (QImaging) through a low-magnification objective (Zeiss 5X/0.16NA). Movies were collected using software custom-written in Labview (National Instruments). Regions of reflectance change were referenced to an image acquired under green illumination.

Two-photon imaging was conducted on the same microscope under the control of the ScanImage software package (V. Iyer, Janelia Farms). All calcium imaging data was collected by two-photon microscopy except for those in *Figure 4*. Scanning during awake conditions was performed at 30 fps using a Chameleon Ultra II laser (Coherent) tuned to 920 nm, precompensated for group velocity dispersion and focused through a 20x/1.0NA water immersion lens (Zeiss). Aquasonic clear ultrasound gel was used for the immersion medium. Emitted light was collected with an HQ535/50 filter (Chroma) and GaAsP photomultiplier tubes (Hamamatsu Photonics). Apical tufts in layer 1 were imaged at depths of 40–80 μm from the pial surface (1.5 x digital zoom in ScanImage which yielded a 433x433 μm field of view, 512x512 pixels).

## SCAPE imaging

High-speed volumetric imaging was performed using a custom SCAPE microscope as previously described, including for dendritic tufts (*Bouchard et al., 2015*; *Hillman et al., 2018*; *Voleti et al., 2019*). Briefly, the cortex was illuminated with an oblique light sheet through a Olympus XLUMPLFLN 20XW 1.0 NA water immersion objective with a 2 mm working distance. Fluorescence excited by this sheet (extending in the *y-z'* direction) was collected by the same objective lens. A galvanometer mirror in the system was positioned to both cause the oblique light sheet to scan from side to side across the sample (in the *x* direction) but also to de-scan returning fluorescence light. This optical path results in an intermediate, de-scanned oblique image plane that is stationary yet always co-aligned with the plane in the sample that is being illuminated by the scanning light sheet. Image rotation optics and a fast sCMOS camera (Andor Zyla 4.2+) were then focused to capture these *y-z'* images (750x200 pixels) at >1000 frames per second as the sheet was repeatedly scanned across the cortex in the *x* direction. All other system parts, including the objective and sample stage, were stationary during high-speed 3D image acquisition. Data were reformed into a 3D volume by stacking successive *y-z'* planes according to the scanning mirror's *x* position and de-skewing to correct for the oblique sheet angle. This rotation of the image volume is responsible for its rectangular appearance despite the camera's square frames. The resulting volumes were large enough to encompass many GCaMP6f-labeled tufts in barrel cortex.

In this study, the stationary objective lens in SCAPE was configured on a manual rotation mount and set to 20°–30° away from the standard upright configuration, so the optical axis was perpendicular to the cranial window to achieve optimal performance without tilting the head of the animal. A 488 nm laser (Coherent OBIS) was used for excitation (<10 mW at the sample) with a 500 nm long-pass filter in the emission path. To achieve optimal spatiotemporal resolution and volume rate, the sample was imaged with an *x*-direction scanning step of 3 μm over a 300×1050 × 234 μm field of view (*x-y-z*, 3.0×1.40 × 1.17 μm per voxel, 100x750 x 200 voxels) at 10 volumes per second (VPS). Our imaging involves no special practical considerations or limitations of field of view or resolution, beyond the usual imaging goal of maximizing FOV while maintaining sufficient resolution to discern structures of interest (dendrites).

## Analysis

Two-photon movies were motion corrected using the NormCorre package (*Pnevmatikakis and Giovannucci, 2017*) in MATLAB or during segmentation using CalmAn. Spatial and temporal components for individual tufts imaged by two-photon and SCAPE were segmented using CalmAn v1.8.3, which employs large-scale sparse non-negative matrix factorization (*Giovannucci et al., 2019*; *Pnevmatikakis et al., 2016*). CalmAn inherently corrects for background signal. All further analyses used custom-written routines implemented in MATLAB. Spatial components with tuft structural characteristics were identified and analyzed, while neuropil components were discarded.

To quantify a tuft's response to stimuli, the mean stimulus-aligned ΔF/F was computed across all CS+ or CS- trials and corrected by the mean ΔF/F of the second before the trial. Probability of transients was obtained by taking each trial's ΔF/F in the first 1.5 seconds following either the CS+ or CS- and fitting these data with a univariate mixture of two Normal distributions: $(1 - p)N(\mu_1, \sigma_1)+pN(\mu_2, \sigma_2)$. The smaller Normal reflects the distribution of failures, and the larger Normal the distribution of transient amplitudes following the stimulus. The parameter $p$ captures the probability of transients.

From these data, a selectivity index (SI) was defined as $(F_{CS+} - F_{CS-}) / (F_{CS+} + F_{CS-})$, in which $F_{CS+}$ and $F_{CS-}$ are the mean stimulus-aligned amplitudes (ΔF/F) to the CS+ and CS- within the first 1.5 s, respectively. This yielded values that range from –1 (exclusively CS- responsive) to 1 (exclusively CS+ responsive). Neural discriminability was defined as $d' = |F_{CS+} - F_{CS-}| / \sqrt{((\sigma^2_{CS+} + \sigma^2_{CS-})/2)}$ where $\sigma^2_{CS+}$ is the variance of the response amplitudes in $F_{CS+}$ and $\sigma^2_{CS-}$ is the variance of the response amplitudes in $F_{CS-}$.

For longitudinal analysis, tufts were categorized as stimulus responsive if they met two criteria: (1) Across all trials, the mean ΔF/F 1.5 s before and 1.5 s after the stimulus were significantly different according to the Wilcoxon rank sum test, for either the CS+ or CS-, and (2) the average response amplitude for that stimulus was greater than 0.04 ΔF/F. Tufts with a significant response to only one stimulus were categorized as highly selective and their |SI| was set to 1. To classify tufts as behaviorally modulated, the mean ΔF/F of the first 1.5 s after the stimulus was computed for false alarm and correct rejection trials and compared with a rank sum test. Only sessions with at least 12 false alarm trials were used for this analysis. If the two distributions were significantly different, the tuft was classified as behaviorally modulated.

Custom MATLAB software was used to compute the median whisker angle, and whisking amplitude was computed as described previously (*Petty et al., 2021*). The median angle was bandpass filtered from 4 to 30 Hz and passed through a Hilbert transform to calculate phase. We defined the upper and lower envelopes of the unfiltered median whisking angle as the points in the whisk cycle where phase equaled 0 (most protracted) or π (most retracted), respectively. Whisking amplitude was defined as the difference between these two envelopes. Periods of whisking were defined as times where whisking amplitude exceeded 20% of maximum for at least 250ms. Periods of time where amplitude exceeded this threshold for less than 250 ms were considered ambiguous and excluded from analysis of whisking versus quiescence. The whisking-triggered average for each tuft was computed by aligning the calcium signal to the start times of whisking periods during inter-trial intervals (2–8 s after stimulus delivery).

For the linear regression analysis, we excerpted the calcium timeseries 2 s before and 6 s after each stimulus onset. The whisking amplitude signal was frame aligned to the calcium signal according to the lag of the calcium-whisking cross-correlation peak for each tuft. Whisking amplitude was then normalized to the max, yielding values that ranged from 0 to 1. The stimulus predictor variable was a binary vector with an 800 ms 'on' period (24 frames) centered at the stimulus time. The timing of the stimulus variable was then aligned to the calcium signal according to the latency of peak of the mean ΔF/F of the first 1.5 s relative to the stimulus. The lick predictor variable was a binary vector with 'on' periods denoting lick bouts. Lick bouts were defined as periods of time where the mouse elicited at least 2 licks, with a maximum gap of 200ms, and therefore had variable lengths.

For support vector machine (SVM) analysis, the mean ΔF/F was computed for a pre-stimulus epoch (1 s immediately preceding the stimulus, used as a negative control) and a post-stimulus epoch (0.1–1.1 s after the stimulus) for each trial. Binary SVMs were trained separately for each epoch using the MATLAB function fitcsvm. For each iteration, 75% of trials were randomly chosen to train the SVM, and decoder performance was tested on the remaining 25% of trials. Decoder performance for each session was averaged across 10 iterations.

All statistical tests were two-sided. T-tests were used for Normally distributed data. Otherwise non-parametric tests were applied.

## Acknowledgements

We thank Venkatakaushik Voleti for help with the design, construction, and alignment of the SCAPE microscope; Dan Kato, Georgia Pierce, and Jung Park for help with pilot experiments; Eftychios Pnevmatikakis and Johannes Friedrich for advice on dendrite segmentation; and Larry Abbott, Stefano Fusi, Ashok Litwin-Kumar, Chris Rodgers, Georgia Pierce, Gordon Petty, and Dan Kato for comments on the manuscript. Funding was provided by a Wellcome Trust Discovery Award, an Academy of

Medical Sciences Professorship, NIH/NINDS R01 NS069679, and NIH/NINDS R01 NS094659 (RMB); a Kavli Institute for Brain Science Postdoctoral Fellowship (SEB); NIH/NINDS/NIMH/BRAIN U01 NS094296, UF1 NS108213, U19 NS104649, and RF1 MH114276 (EMCH).

## Additional information

### Competing interests

Elizabeth MC Hillman: A patent related to the SCAPE microscopy technique used in Figure 4 was issued on December 31st 2013, and the author has licensed the technology. The other authors declare that no competing interests exist.

### Funding

| Funder | Grant reference number | Author |
|---|---|---|
| Wellcome Trust | 225210/Z/22/Z | Randy M Bruno |
| Academy of Medical Sciences | APR6\1007 | Randy M Bruno |
| National Institute of Neurological Disorders and Stroke | R01 NS069679 | Randy M Bruno |
| National Institute of Neurological Disorders and Stroke | R01 NS094659 | Randy M Bruno |
| Kavli Institute for Brain Science | | Sam E Benezra |
| National Institutes of Health | U01 NS094296 | Elizabeth MC Hillman |
| National Institutes of Health | UF1 NS108213 | Elizabeth MC Hillman |
| National Institutes of Health | U19 NS104649 | Elizabeth MC Hillman |
| National Institutes of Health | RF1 671 MH114276 | Elizabeth MC Hillman |

The funders had no role in study design, data collection and interpretation, or the decision to submit the work for publication. For the purpose of Open Access, the authors have applied a CC BY public copyright license to any Author Accepted Manuscript version arising from this submission.

### Author contributions

Sam E Benezra, Conceptualization, Data curation, Formal analysis, Investigation, Visualization, Methodology, Writing – original draft, Writing – review and editing; Kripa B Patel, Citlali Perez Campos, Methodology; Elizabeth MC Hillman, Conceptualization, Formal analysis, Supervision, Funding acquisition, Methodology; Randy M Bruno, Conceptualization, Formal analysis, Supervision, Funding acquisition, Methodology, Writing – original draft, Writing – review and editing

### Author ORCIDs

Elizabeth MC Hillman ⓘ https://orcid.org/0000-0001-5511-1451
Randy M Bruno ⓘ https://orcid.org/0000-0002-5122-4632

### Ethics

All experiments complied with the NIH Guide for the Care and Use of Laboratory Animals and were approved by the Institutional Animal Care and Use Committee of Columbia University (protocol number AC AABP0555).

Reviewer #1 (Public review): https://doi.org/10.7554/eLife.98349.3.sa1

Reviewer #2 (Public review): https://doi.org/10.7554/eLife.98349.3.sa2
Author response https://doi.org/10.7554/eLife.98349.3.sa3

## Additional files

**Supplementary files**
MDAR checklist

**Data availability**
Data from this study are available for download on Dryad: https://doi.org/10.5061/dryad.v6wwpzh5t.

The following dataset was generated:

| Author(s) | Year | Dataset title | Dataset URL | Database and Identifier |
|---|---|---|---|---|
| Benezra SE, Patel KB, Campos CP, Hillman EMC, Bruno RM | 2024 | Data from: Learning enhances behaviorally relevant representations in apical dendrites | https://doi.org/10.5061/dryad.v6wwpzh5t | Dryad Digital Repository, 10.5061/dryad.v6wwpzh5t |

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
