## [Editor Report · eLife Assessment]

This **important** study uses calcium imaging to show an increase in the selectivity of the sensory-evoked response in the apical dendritic tuft of layer 5 barrel cortex neurons as mice learn a whisker-dependent discrimination task. The evidence supporting the conclusions is **compelling**, and this work will be of great interest to neuroscientists working on reward-based learning and sensory processing.

---

## [Referee Report · Reviewer #1 (Public review)]

What neurophysiological changes support the learning of new sensorimotor transformations is a key question in neuroscience. Many studies have attempted to answer this question at the neuronal population level - with varying degrees of success - but few, if any, have studied the change in activity of the apical dendrites of layer 5 cortical neurons. Neurons in the layer 5 of the sensory cortex appear to play a key role in sensorimotor transformations, showing important decision and reward-related signals, and being the main source of cortical and subcortical projections from the cortex. In particular, pyramidal track (PT) neurons project directly to subcortical regions related to motor activity, such as the striatum and brainstem, and could initiate rapid motor action in response to given sensory inputs. Additionally, layer 5 cortical neurons have large apical dendrites that extend to layer 1 where different neuromodulatory and long-range inputs converge, providing motor and contextual information that could be used to modulate layer 5 neurons output and/or to establish the synaptic plasticity required for learning a new association.

In this study, the authors aimed to test whether the learning of a new sensorimotor transformation could be supported by a change in the evoked response of the apical dendrites of layer 5 neurons in the mouse whisker primary somatosensory cortex. To do this, they performed longitudinal functional calcium imaging of the apical dendrites of layer 5 neurons while mice learned to discriminate between two multiwhiskers stimuli. The authors used a simple conditioning task in which one whisker stimulus (upward or backward air puff, CS+) is associated with reward after a short delay, while the other whisker stimulus (CS-) is not. They found that task learning (measured by the probability of anticipatory licking just after the CS+) was not associated with a significant change of the average population response evoked by the CS+ or the CS-, nor change in the average population selectivity. However, when considering individual dendritic tufts, they found interesting changes in selectivity, with approximately equal numbers of dendrites becoming more selective for CS+ and dendrites becoming more selective for CS-.

One of the major challenges when assessing changes in neural representation during the learning of such Go/NoGo tasks is that the movements and rewards themselves may elicit strong neural responses that may be a confounding factor, that is, inexperienced mice do not lick in response to the CS+, while trained mice do. In this study, the authors addressed this issue in three ways: first, they carefully monitor the orofacial movements of mice and show that task learning is not associated with changes in evoked whisker movements. Second, they show that whisking or licking evokes very little activity in the dendritic tufts compared to whisker stimuli (CS+ and CS-). Finally, the authors introduced into the design of their task a post-conditioning session after the last conditioning session during which the CS+ and the CS- are presented but no reward is delivered. During this post-session, the mice gradually stopped licking in response to the CS+. A better design might have been to perform the pre-conditioning and post-conditioning sessions in non-water-restricted, unmotivated mice to completely exclude any lick response, but the fact that the change in selectivity persists after the mice stopped licking in the last blocks of the post-conditioning session (in mice relying only on their whiskers to perform the task) is convincing.

The clever task design and careful data analysis provide compelling evidence that learning this whisker discrimination task does not result in a massive change in sensory representation in the apical dendritic tufts of layer 5 neurons in the primary somatosensory cortex on average. Nevertheless, individual dendritic tufts do increase their selectivity for one or the other sensory stimulus, likely enhancing the ability of S1 neurons to accurately discriminate the two stimuli and trigger the appropriate motor response (to lick or not to lick).

One limitation of the present study is the lack of evidence for the necessity of the primary somatosensory cortex in the learning and execution of the task. As the authors have strongly emphasized in their previous publications, the primary somatosensory cortex may not be necessary for the learning and execution of simple whisker detection tasks, especially when the stimulus is very salient. Although this new task requires the discrimination between two whisker stimuli, the simplicity and salience of the whisker stimuli used could make this task cortex independent. Especially when considering that some mice seem to not rely entirely on their whiskers to execute the task.

Nevertheless, this is an important result that shows for the first-time changes in the selectivity to sensory stimuli at the level of individual apical dendritic tufts in correlation with the learning of a discrimination task. This study sheds new light on the cortical cellular substrates of reward-based learning, and opens interesting perspectives for future research in this area. In future studies, it will be important to determine whether the change in selectivity of dendritic calcium spikes is causally involved in the learning the task or whether it simply correlates with learning, as a consequence of changes in synaptic inputs caused by reward. The dendritic calcium spikes may be involved in the establishment of synaptic plasticity required for learning and impact the output of layer 5 pyramidal neurons to trigger the appropriate motor response. It would be important also to study the changes in selectivity in the apical dendrite of the identified projection neurons.

Comments on revisions:

The authors have addressed all my questions. I have no further recommendations.

---

## [Referee Report · Reviewer #2 (Public review)]

Summary:

The authors did not find an increased representation of CS+ throughout reinforcement learning in the tuft dendrites of Rbp4-positive neurons from layer 5B of the barrel cortex, as previously reported for soma from layer 2/3 of the visual cortex.

Alternatively, the authors observed an increased selectivity to both stimuli (CS+ and CS-) during reinforcement learning. This feature (1) was not present in repeated exposures (without reinforcement), (2) was not explained by animal's behaviour (choice, licking and whisking) and (3) was long-lasting, being present even when the mice disengaged from the task.

Importantly, increased selectivity was correlated with learning (% correct choices), and neural discriminability between stimuli increased with learning.

In conclusion, the authors show that tuft dendrites from layer 5B of the barrel cortex increase the representation of conditioned (CS+) and unconditioned stimuli (CS-) applied to the whiskers, during reinforcement learning.

Strengths:

The results presented are very consistent throughout the entire study, and therefore very convincing:

(1) The results observed are very similar using two different imaging techniques (using 2-photon -planar imaging- and SCAPE - volumetric imaging). Fig. 3 and Fig.4 respectively.

(2) The results are similar using "different groups" of tuft dendrites for the analysis (e.g. initially unresponsive and responsive pre- and post-learning). Fig. 5.

(3) The results are similar from a specific set of trials (with the same sensory input, but different choices). Fig.7.

(4) Additionally, the selectivity of tuft dendrites from layer 5B of the barrel cortex was higher in the mice that exclusively used the whisker to respond to the stimuli (CS+ and CS-).

The results presented are controlled against a group of mice that received the same stimuli presentation, except the reinforcement (reward).

Additionally, the behaviour outputs, such as choice, whisking and licking could not account for the results observed.

Although there are no causal experiments, the correlation between selectivity and learning (% of correct choices), as well as the increased neural discriminability with learning, but not in repeated exposure, are very convincing.

Weaknesses:

The biggest weakness is the absence of causality experiments. Although inhibiting specifically tuft dendritic activity in layer 1 from layer 5 pyramidal neurons is very challenging, tuft dendritic activity in layer 1 could be silenced through optogenetic experiments as in Abs et al. 2018. By manipulating NDNF-positive neurons the authors could specifically modify tuft dendritic activity in the barrel cortex during CS presentations, and test if silencing tuft dendritic activity in layer 1 would lead to the lack of selectivity and an impairment of reinforcement learning. Additionally, this experiment will test if the selectivity observed during reinforcement learning is due to changes in the local network, namely changes in local synaptic connectivity, or solely due to changes in the long-range inputs.

---

## [Author Response]

The following is the authors’ response to the original reviews.

**Reviewer #1 (Public Review):**
What neurophysiological changes support the learning of new sensorimotor transformations is a key question in neuroscience. Many studies have attempted to answer this question at the neuronal population level - with varying degrees of success - but few, if any, have studied the change in activity of the apical dendrites of layer 5 cortical neurons. Neurons in layer 5 of the sensory cortex appear to play a key role in sensorimotor transformations, showing important decision and reward-related signals, and being the main source of cortical and subcortical projections from the cortex. In particular, pyramidal track (PT) neurons project directly to subcortical regions related to motor activity, such as the striatum and brainstem, and could initiate rapid motor action in response to given sensory inputs. Additionally, layer 5 cortical neurons have large apical dendrites that extend to layer 1 where different neuromodulatory and long-range inputs converge, providing motor and contextual information that could be used to modulate layer 5 neurons output and/or to establish the synaptic plasticity required for learning a new association.In this study, the authors aimed to test whether the learning of a new sensorimotor transformation could be supported by a change in the evoked response of the apical dendrites of layer 5 neurons in the mouse whisker primary somatosensory cortex. To do this, they performed longitudinal functional calcium imaging of the apical dendrites of layer 5 neurons while mice learned to discriminate between two multi-whisker stimuli. The authors used a simple conditioning task in which one whisker stimulus (upward or backward air pu , CS+) is associated with a reward after a short delay, while the other whisker stimulus (CS-) is not. They found that task learning (measured by the probability of anticipatory licking just after the CS+) was not associated with a significant change in the average population response evoked by the CS+ or the CS-, nor a change in the average population selectivity. However, when considering individual dendritic tufts, they found interesting changes in selectivity, with approximately equal numbers of dendrites becoming more selective for CS+ and dendrites becoming more selective for CS-.One of the major challenges when assessing changes in neural representation during the learning of such Go/NoGo tasks is that the movements and rewards themselves may elicit strong neural responses that may be a confounding factor, that is, inexperienced mice do not lick in response to the CS+, while trained mice do. In this study, the authors addressed this issue in three ways: first, they carefully monitored the orofacial movements of mice and showed that task learning is not associated with changes in evoked whisker movements. Second, they show that whisking or licking evokes very little activity in the dendritic tufts compared to whisker stimuli (CS+ and CS-). Finally, the authors introduced into the design of their task a post-conditioning session after the last conditioning session during which the CS+ and the CS- are presented but no reward is delivered. During this post-session, the mice gradually stopped licking in response to the CS+. A better design might have been to perform the pre-conditioning and post-conditioning sessions in nonwater-restricted, unmotivated mice to completely exclude any lick response, but the fact that the change in selectivity persists after the mice stopped licking in the last blocks of the post-conditioning session (in mice relying only on their whiskers to perform the task) is convincing.The clever task design and careful data analysis provide compelling evidence that learning this whisker discrimination task does not result in a massive change in sensory representation in the apical dendritic tufts of layer 5 neurons in the primary somatosensory cortex on average. Nevertheless, individual dendritic tufts do increase their selectivity for one or the other sensory stimulus, likely enhancing the ability of S1 neurons to accurately discriminate the two stimuli and trigger the appropriate motor response (to lick or not to lick).One limitation of the present study is the lack of evidence for the necessity of the primary somatosensory cortex in the learning and execution of the task. As the authors have strongly emphasized in their previous publications, the primary somatosensory cortex may not be necessary for the learning and execution of simple whisker detection tasks, especially when the stimulus is very salient. Although this new task requires the discrimination between two whisker stimuli, the simplicity and salience of the whisker stimuli used could make this task cortex-independent. Especially when considering that some mice seem to not rely entirely on their whiskers to execute the task.Nevertheless, this is an important result that shows for the first time changes in the selectivity to sensory stimuli at the level of individual apical dendritic tufts in correlation with the learning of a discrimination task. This study sheds new light on the cortical cellular substrates of reward-based learning and opens interesting perspectives for future research in this area. In future studies, it will be important to determine whether the change in selectivity of dendritic calcium spikes is causally involved in the learning of the task or whether it simply correlates with learning, as a consequence of changes in synaptic inputs caused by reward. The dendritic calcium spikes may be involved in the establishment of synaptic plasticity required for learning and impact the output of layer 5 pyramidal neurons to trigger the appropriate motor response. It would be important also to study the changes in selectivity in the apical dendrite of the identified projection neurons.
**Reviewer #2 (Public Review):**
Summary:The authors did not find an increased representation of CS+ throughout reinforcement learning in the tuft dendrites of Rbp4-positive neurons from layer 5B of the barrel cortex, as previously reported for soma from layer 2/3 of the visual cortex.Alternatively, the authors observed an increased selectivity to both stimuli (CS+ and CS-) during reinforcement learning. This feature:(1) was not present in repeated exposures (without reinforcement),(2) was not explained by the animal's behaviour (choice, licking, and whisking), and(3) was long-lasting, being present even when the mice disengaged from the task.Importantly, increased selectivity was correlated with learning (% correct choices), and neural discriminability between stimuli increased with learning.In conclusion, the authors show that tuft dendrites from layer 5B of the barrel cortex increase the representation of conditioned (CS+) and unconditioned stimuli (CS-) applied to the whiskers, during reinforcement learning.Strengths:The results presented are very consistent throughout the entire study, and therefore very convincing:(1) The results observed are very similar using two different imaging techniques (2-photon planar imaging- and SCAPE-volumetric imaging). Figure 3 and Figure 4 respectively.(2) The results are similar using "different groups" of tuft dendrites for the analysis (e.g. initially unresponsive and responsive pre- and post-learning). Figure 5.(3) The results are similar from a specific set of trials (with the same sensory input, but di erent choices). Figure 7.(4) Additionally, the selectivity of tuft dendrites from layer 5B of the barrel cortex was higher in the mice that exclusively used the whisker to respond to the stimuli (CS+ and CS-). The results presented are controlled against a group of mice that received the same stimuli presentation, except for the reinforcement (reward).Additionally, the behaviour outputs, such as choice, whisking, and licking could not account for the results observed.Although there are no causal experiments, the correlation between selectivity and learning (percentage of correct choices), as well as the increased neural discriminability with learning, but not in repeated exposure, are very convincing.Weaknesses:The biggest weakness is the absence of causality experiments. Although inhibiting specifically tuft dendritic activity in layer 1 from layer 5 pyramidal neurons is very challenging, tuft dendritic activity in layer 1 could be silenced through optogenetic experiments as in Abs et al. 2018. By manipulating NDNF-positive neurons the authors could specifically modify tuft dendritic activity in the barrel cortex during CS presentations, and test if silencing tuft dendritic activity in layer 1 would lead to the lack of selectivity and an impairment of reinforcement learning. Additionally, this experiment will test if the selectivity observed during reinforcement learning is due to changes in the local network, namely changes in local synaptic connectivity, or solely due to changes in the long-range inputs.

We agree that such causal manipulations are a logical next step. Such manipulations are unfortunately not specific to layer 5 apicals, so the results would be difficult to interpret. We now discuss the challenge of such manipulations in the Discussion section.

**Recommendations for the authors:**

**Reviewer #1 (Recommendations For The Authors):**
Overall, the study is solid and the article is well and clearly written. I have no suggestion for other experiments that would fall within the scope of this article. I would like only to suggest some additional analyses and clarifications in the writing.Additional analyses:Obviously, the main confounding factor in this type of data comes from the acquired motor response which follows - with a short latency - the sensory stimulus. This is particularly problematic for functional calcium imaging which has very low temporal resolution. The authors have addressed this question to some extent by showing that motor-evoked activity does not account for the change in selectivity acquired with learning and through the use of a post-conditioning session during which no reward was delivered. Figures 8C-D show that mice gradually stop licking in response to CS+ in this session and that the distribution of the selectivity index remains similar in these last blocks. Perhaps a more convincing analysis would be to simply select Miss and Correct rejection trials in which mice did not lick in response to the CS+ and CS-, respectively. Ideally, if the number of trials is sufficient, one could even select trials devoid of any evoked movement (no licking and no whisking).

We agree it would be interesting to compare Miss and Correct rejection trials to further rule out effects of a motor response, but there were never enough Miss trials to conduct such an analysis. Even in very early learning, there are few Miss trials (see Figure 1, session 2). We found that in early learning, animals would lick in most trials. Then, over the course of conditioning, they would learn to withhold licks during CS- presentation. Thus, we were able to examine Hits, Correct rejections, and False alarms (Figure 7), but not Miss trials. We have added text suggesting a future experiment in which the stimulus strengths are substantially reduced to drastically increase the error rates.

The fact that changes in selectivity occur in both directions overall is really interesting. However, in the way the data are presented currently, one may wonder about mice/field of view vs single cell effect. i.e., do di erent dendritic tufts in the same field of view show opposite changes in selectivity? If we were to replot Figure 3A for a single mouse, would we obtain the same picture?

We appreciate this very good suggestion and have added scatter plots and selectivity index histograms for individual conditioned animals in Supplementary figure 2. These data demonstrate that different dendritic tufts in the same field of view exhibit opposite changes in selectivity.

The authors point out that they observed no change in the mean response or selectivity during learning, but did find changes in selectivity at the level of individual dendritic tufts. This suggests that, at the population level, the ability to discriminate between the two stimuli should improve. A possible complementary analysis would be to show that the ability to decode stimulus identity from dendritic tuft population activity increases with learning.

Given the substantial change in individual tuft selectivity and that the tuft events occur are not rare, the population result is guaranteed. If individual tufts increase selectivity, the population will also increase its selectivity on a trial-by-trial basis. We have nevertheless included a new supplementary figure with a population analysis using SVMs to demonstrate this.

Clarification:The authors should make it clear from the beginning that mice are still water-restricted during the post-conditioning session and actually do keep licking for many CS+ trials. Therefore, this session is not devoid of motor response.

We have clarified this in the text.

Did mice in the repeated exposure condition receive any reward during the recording sessions? If so when were rewards delivered?

We previously described in the Methods that these mice received water in their home cage, but we now additionally clarify this in the Results section.

Minor:Figure 2Aii, the labels of the Alpha and Betta barrels should be swapped.

Fixed

Line 218: I believe this sentence should read "Using SCAPE microscopy, ...".

Corrected.

Line 665: 'Reconstruction from 50' does that refer to the single cell reconstruction on the left panel?

Yes – Clarified in legend

**Reviewer #2 (Recommendations For The Authors):**
Minor suggestions:The 'summary' should mention from which brain area the results were acquired. Otherwise, it is misleading, giving the idea that the results described a generic feature, which is still unknown.

Added to the text.

Please correct sentence 219: "SCAPE microscopy, we image tuft activity of additional mice..."

Added to the text.

In the same sentence (219) it would be good to provide the number of additional mice imaged (2).

Added to the text.

Regarding Supplementary Figure 1, it would be interesting to correlate the second peak after reward and learning rate, to provide further support to the sentences 109 to 113.

We agree this would be interesting to examine, but only four animals exhibited this second peak, which is too small of a sample to observe a meaningful correlation. We now clarify this in the text.

In Figure 3, why not present the correlation between 'neural discriminability' and % of correct choices?

We appreciate the suggestion and have added this plot to Figure 3.

The 'results' section will benefit tremendously if the authors consistently indicate the figures to which the results are being described, or 'data not shown' if it is the case. To give a few examples:Sentence 108 - "averaged 28% ΔF/F" - From which figure is this result coming from? Sentence 123 - "(p = 0.62, 0.64, respectively)" - comparison not shown, but see Figures 2E and D respectively?Sentence 125 - "(CS+ responsive (...) across all sessions)" - From which figure is this result coming from?Sentence 130 - "during pre-conditioning (p=0.66) or post-conditioning sessions (p=0.44) - From which figure?Sentence 154 - "(Pre: p=0.20; last rewarded: p=0.43; Post: p=0.64, sign-rank test)" - From which figure?Sentence 175 - "(-0.049, -0.001, and 0.003)" - From which figure? Please show the graph that shows that the mean SI is not different. It can be supplementary. The distribution of SI will be strengthened by it.

We added this plot to supplementary figure 2.

Sentence 244 - "(conditioned: 458/603; repeated exposure: 334/457) - From Figure 5E.Sentence 256 - "(p=0.04, 2-sample t-test comparison mice) - From Figure 5B. Sentence 258 - "(p=0.03, paired t-test) - from Figure 5B Sentences 370 to 378 - No reference to the figure.The 'discussion' section (sentences 459 to 494) refers to the differences between the current and previous studies (references 1,3,5), namely soma vs. dendrites and layer 2/3 vs. layer 5. However, it should also mention the difference between the nature of the stimuli and the brain area recorded (visual cortex vs. barrel cortex).

We have addressed these issues in the text.